

# Riparian evapotranspiration shapes stream flow dynamics and water budgets in a Mediterranean catchment

**Authors:** Anna Lupon[1], José L. J. Ledesma[2], Susana Bernal[3,4]

**Affiliations:**

[1] Department of Forest Ecology and Management, Swedish University of Agricultural Sciences (SLU). Skogsmarksgränd, 901 83 Umeå, Sweden.

[2] Department of Aquatic Sciences and Assessment, Swedish University of Agricultural Sciences (SLU). Lennart Hjelms väg 9, SE, 750 07 Uppsala, Sweden.

[3] Departament de Biologia Evolutiva, Ecologia i Ciències Ambientals, Universitat de Barcelona. Av. Diagonal 643, 08028 Barcelona, Spain.

[4] Integrative Freshwater Ecology Group, Center for Advanced Studies of Blanes (CEAB-CSIC). Accés a la Cala Sant Francesc 14, 17300 Blanes, Spain.

*Correspondence to:* Anna Lupon (anna.lupon@slu.se)



## Abstract

Riparian trees can regulate stream flow dynamics and water budgets by taking up large amounts of water from both soil and groundwater compartments. However, their role has not been fully recognized in the hydrologic literature and the catchment modeling community. In this study, we explored the influence of riparian evapotranspiration (ET) on stream flow by simulating daily stream exports from three nested Mediterranean sub-catchments, both including and excluding the riparian compartment in the structure of the PERSiST rainfall-runoff model. The model goodness of fit significantly improved with the inclusion of the riparian compartment, especially during the vegetative period when, according to our simulations, riparian ET reduced mean daily stream flow by 26%. Moreover, sensitivity analyses suggested that riparian ET was a significant hydrological process contributing to stream flow recession in summer. At the catchment scale, simulated riparian ET accounted for 7% of annual water depletions, its contribution being especially noticeable during summer (8–19%). Simulations considering future climate change scenarios suggest that longer vegetative periods would result in higher contribution of riparian ET to annual water budgets. Annual increases in riparian ET ranged between 2 and 13% from the most conservative to the most extreme drought scenarios. Overall, our results highlight that a good assessment of riparian ET is essential for understanding catchment hydrology and stream flow dynamics in Mediterranean regions. Thus, the inclusion of the riparian compartment in hydrological models is strongly recommended in order to establish proper management strategies in water-limited regions.

**Keywords:** PERSiST model, riparian evapotranspiration, water resources, stream flow, Mediterranean regions, climate change, aridity index.





## 1 Introduction

35 

Precipitation and upland tree evapotranspiration (ET) are considered the two most important components controlling annual water budgets in catchment hydrology. This conceptualization is supported by the fact that, in most regions, landscape units other than uplands (e.g. riparian zones) occupy a small percentage of the catchment area (< 3%) (Tockner and Stanford, 2002). However, empirical studies have shown that

40 riparian ET can influence stream flow dynamics by lowering groundwater levels and increasing groundwater residence times (Bernal et al., 2004; Burt et al., 2002). In water limited regions, water demand by riparian trees can influence the exchange of water between terrestrial and fluvial ecosystems by disconnecting saturated soils from streams and promoting the displacement of stream water towards the riparian zone (Butturini et al., 2003; Lupon et al., 2016; Rassam et al., 2006). Altogether, these studies

45 suggest that hydrological processes occurring in the riparian zone, specifically those induced by riparian ET, can be critical to understand stream flow dynamics in regions potentially suffering from water scarcity. Yet, most of these studies have been conducted at plot or reach scales and there are still few exercises assessing the influence of riparian ET on stream flow and water exports at the catchment scale.

Riparian trees can play an important role in catchment water budgets because their water requirements

50 are generally high compared to upland tree species (Baldocchi and Ryu, 2011; Doody and Benyon, 2011). Yet, the contribution of riparian ET to catchment annual water budgets varies widely among biomes (from 0% to > 30%) depending on the amount of water available for vegetation (Dahm et al., 2002; Cadol et al., 2012; Contreras et al., 2011). In tropical systems, for instance, soil water content is usually high in both upland and riparian zones, and hence, these two compartments show similar ET rates (2–5 mm d$^{-1}$; Cadol

55 et al., 2012; da Rocha et al., 2004). Conversely, in arid systems, riparian zones stay relatively wet compared to upland areas and can support ET rates between 1 and 7 mm d$^{-1}$, as much as one order of magnitude higher than those in the surrounding upland (0.1–0.4 mm d$^{-1}$; Dahm et al., 2002; Kurc and Small, 2004). These differences among biomes suggest that the potential of riparian forests to shape stream flow dynamics and water budgets likely increases with increasing water scarcity.

60 Mediterranean catchments are unique natural laboratories for evaluating the influence of riparian ET on stream and catchment hydrology as well as to test the response of riparian ET to changes in climatic



drivers, namely temperature and precipitation. Mediterranean regions exhibit marked seasonal patterns in both hydrology and vegetative activity, and they hold an intermediate position in the climatic gradient, which makes them especially vulnerable to future changes in climate (IPCC, 2013). Previous studies have

shown that capturing abrupt changes in groundwater tables associated with summer riparian ET are essential to predict daily and seasonal shifts in stream flow in Mediterranean areas (Lupon et al., 2016; Medici et al., 2008). Thus, a better understanding of the implications of these hydrological processes on catchment water budgets and on water availability for both in- and off-stream uses is needed. This information is critical to assess the hydrological management of Mediterranean forested catchments as

well as to achieve feasible predictions of hydrological and ecological responses to future climate.

The aim of this study was to explore the role of riparian ET on regulating present and future stream flow dynamics and catchment water exports in a Mediterranean forested headwater catchment on a seasonal and annual basis. To do so, we used the rainfall-runoff model PERSiST (Precipitation, Evapotranspiration and Runoff Simulator for Solute Transport; Futter et al., 2014) to reproduce the observed stream

hydrographs at three nested sub-catchments along which the area covered by riparian forests varied from 0 to 10%. We expected that the influence of riparian ET on stream flow dynamics and catchment water exports will be magnified during late spring and summer, when hydrological connectivity between uplands and the stream network is low. Moreover, we simulated different climate scenarios for the region in order to explore changes in the relative contribution of riparian ET to total catchment water budgets

with increasing drying.

## 2  Study site

The Font del Regàs catchment is located in the Montseny Natural Park, NE Spain (41°50'N, 2°30'E). The climate is subhumid Mediterranean, with mild winters, wet springs, and dry summers. Annual precipitation is 925 ± 151 mm (mean ± SD), less than 1% falling as snow. Mean annual temperature

averages 12.1 ± 2.5 °C (period 1940–2000, Catalan Metereologic Service).

Total catchment area is 14.2 km$^2$ and its altitude ranges from 500 to 1500 m above the sea level (a.s.l.) (Figure 1). The geology is dominated by biotitic granite and the topography includes steep slopes (28%)



(Institut Cartografic de Catalunya, 2010). Evergreen oak forests (*Quercus ilex*) cover the lower part of the catchment (54% of the catchment area), whereas the upper part is covered mainly by deciduous European beech (*Fagus sylvatica*) forests and heathlands (38 and 2% of the catchment area, respectively) (Figure 1). Upland soils (pH ~ 6) are sandy, with a 3 cm deep O horizon followed by a 5 to 15 cm deep A horizon. The riparian forest covers 6% of the total catchment area and it is relatively flat (slope < 10%). Riparian width increases from 6 to 28 m along the catchment and the total basal area of riparian trees increases by 12-fold. Black alder (*Alnus glutinosa*), European ash (*Fraxinus excelsior*), black locust (*Robinea pseudoacacia*), and black poplar (*Populus nigra)* are the most abundant tree species in the riparian forest, with a basal area of 14, 4, 3 and 2 $m^2$ $ha^{-1}$, respectively. Riparian soils (pH ~ 7) are sandy-loam, with a 5 cm deep organic layer followed by a 30 cm deep A horizon.

For this study, we selected three nested sub-catchments along a 5.6 km stretch of the Font del Regàs stream (Figure 1, Table 1). The upstream sub-catchment (800–1500 m a.s.l, local drainage area 1.8 $km^2$) was mostly composed by beech forest (93%) and had no riparian forest. Vegetation in the midstream sub-catchment (650–800 m a.s.l., local drainage area 6.74 $km^2$) included both oak (52.5%) and beech (42.5%) forests. The stream at the midstream sub-catchment had a wetted width of 2–3 m and was flanked by a mixed forest (5%, 5–15 m wide) of *Alnus glutinosa* and *Fraxinus excelsior*. The downstream sub-catchment (500–650 m a.s.l., local drainage area 4.42 $km^2$) was mainly covered by oak forest (58%) and, to a lesser extent, by beech forest (32%). The stream at the downstream sub-catchment had a wetted width of 3–3.5 m and was flanked by a well-developed riparian forest (10%, 15–30 m wide) consisting mainly of *Robinea pseudoacacia, Populus nigra,* and *Alnus glutinosa*.

## 3 Materials and methods

### 3.1 The PERSiST model

PERSiST is a conceptual, semi-distributed, bucket-type model that simulates daily catchment water fluxes (Futter et al., 2014). The flexible model framework allows representing the runoff generation process as a specified number of vertically and horizontally interconnected buckets (representing soil layers) within a mosaic of landscape units. In this way, PERSiST allows differentiating the riparian compartment from



other catchment water pools (see below). The riparian water fluxes represented in the model are
subsurface flow and evapotranspiration (Futter et al., 2014).

In the model, rainfall is directed to the stream as overland flow or infiltrated to the upper soil layer.
Vertically, water can move to lower soil layers via infiltration or return to the atmosphere via ET.
Horizontally, soil water moves downhill to other catchment compartments or, alternatively, to the stream.
Water movement is controlled by field capacities, hydrological connectivity, and infiltration-related
parameters. Water lost via ET is landscape unit-specific and is controlled by degree day rates and
threshold temperature parameters. During dry conditions, (i) an ET adjustment parameter can be used at
the soil layer level to limit ET and (ii) a specified fraction of the incoming rainfall can be directly
transported to stream runoff. Rainfall can also be intercepted by the canopy. The magnitude and flashiness
of the simulated flow is also dependent on the catchment area and water velocity-related parameters.
Catchment and landscape unit-specific rain multipliers correct for potential rainfall measurement biases.
Finally, landscape unit-specific parameter values are used to specify the fraction of water moving between
contiguous soil layers and with the stream at every time step.

### 3.2 Model input data and configuration

We used PERSiST to explore the influence of riparian ET on the seasonal variation of stream flow at Font
del Regàs. We calibrated PERSiST for two complete hydrological years (Sep 2010–Aug 2012) at the
outlet of the up-, mid-, and downstream sub-catchments using as input data time series of daily
precipitation (in mm) and mean daily air temperature (in ℃). Both precipitation and temperature were
recorded at 15-min intervals at a meteorological station located at the valley bottom of the catchment
(Figure 1) and converted to daily values for model simulation. At each sub-catchment outlet, stream water
level was monitored at 15-min intervals using a water pressure sensor connected to an automatic sample
(Model 1612, Teledyne Isco) (Figure 1). An empirical relationship between stream flow (in L s$^{-1}$) and
stream water level was obtained using the slug chloride addition method in the field (n = 57, 60, and 61
for up-, mid-, and downstream sub-catchments, respectively; in all cases: $R^2 > 0.97$; Lupon et al., 2016).
Daily accumulated flows considering the total upstream drainage area at the three locations were used for
model calibration. Model simulation was started in January 2010 to have an 8-month warm-up period.



To investigate the importance of riparian ET to the temporal pattern of stream flow at Font del Regàs, we calibrated the model for the three sub-catchment outlets (referred as to "stream sites" hereafter) both including and excluding the riparian compartment in the model structure. The aim was to determine whether riparian ET improved the goodness of fit between observed and simulated stream flow. In the first model configuration (not including a riparian compartment), we used a simple one-compartment approach to represent the catchment area in all three sub-catchments (i.e., a single upland representation). For each sub-catchment, the upland compartment was divided into two landscape units representing evergreen and deciduous forests in appropriate proportions (Table 1), and the soil was divided into three layers representing overland (quick), upper (soil), and lower (groundwater) strata.

The second model configuration consisted on both an upland and a riparian compartment, which was added for the mid- and downstream sub-catchments (5 and 10% area covered by riparian forest, respectively) (Table 1). In this configuration, the upper upland layer communicated downhill with the riparian layer and vertically with the groundwater. The riparian layer also received water inputs from groundwater, which was shared for both upland and riparian compartments. The stream received water inputs from both the riparian compartment and the groundwater. Following knowledge of the area, overland flow was not used in either model configuration or all water entering the overland (quick) bucket percolated to the upper soil layer either in the upland or riparian compartments. ET was simulated from uplands and riparian layers separately. The importance of riparian ET on simulating stream water flow and catchment water budgets was determined by comparing specific statistical metrics of goodness of fit (see below) from the two model configurations (including and excluding the riparian compartment).

### 3.3 Calibration procedure

Model calibration was done manually for all six model instances (3 sub-catchments x 2 model configurations) in order to (i) match ET values reported for the different forest types ("soft calibration") and (ii) optimize a combination of statistical metrics (i.e. model efficiency) that compare simulated and observed flows ("hard calibration"). Note that the instances from the upstream site were equivalent in the two configurations because this sub-catchment had no riparian forest. Manual calibration has been proved as a robust method for obtaining acceptable simulations within the Integrated Catchment (INCA) family





of models, of which PERSiST is the common hydrological model (Cremona et al., 2017; Futter et al., 2014; Ledesma et al., 2012).

For the soft calibration, the parameterization of both upland and riparian ET was adjusted to obtain values of water demand within the ranges reported for evergreen forest (i.e. evergreen oak; 550–650 mm yr$^{-1}$), deciduous forest (i.e. beech; 600–750 mm yr$^{-1}$), and riparian forests (i.e. poplar, alder and ash; 750–1000 mm yr$^{-1}$) at Montseny or nearby (< 50 km) mountains (Àvila et al., 1996; Folch and Ferrer, 2015; Llorens and Domingo, 2007; Sabater and Bernal, 2011). We calibrated the model assuming (i) a higher ET from

evergreen forest than from deciduous and riparian forests during the dormant period and (ii) a higher riparian ET than evergreen and deciduous ET during the vegetative period. The first assumption was based on the premise that deciduous trees cannot transpire during the dormant period, while the second assumption was based on the idea that riparian trees are closer to water sources, and thus, they are not as water limited as upland trees (both evergreen oak and beech) in summer. Other parameterization

requirements during soft calibration included matching reported annual canopy rainfall interception values for similar forest types (Àvila et al., 1996; Terradas, 1984; Terradas and Savé, 1992) and a rainfall correction for south- and north-facing slopes which roughly corresponded to evergreen and deciduous forests, respectively (Piñol et al., 1992).

    For the hard calibration, model parameters related to water fluxes, including the fraction of water

communicating soil layers and time constants (a proxy for hydrological connectivity), were adjusted to optimize the Nash and Sutcliffe (NS) efficiency index (important to fit high flows), the log NS (important to fit low flows), the relative volume differences of observed versus simulated stream flow (RVD) (important to maintain the water balance), and the overall graphical fit between observed and simulated hydrographs. For both NS and log(NS), higher values indicate a better goodness of fit. For RVD, positive

and negative values indicate that the model under- and overestimated the stream flow, respectively. The three indexes (NS, log(NS), and RVD) were estimated for the six model instances for the whole data set as well as for the riparian vegetative (April–October) and dormant (November–March) periods separately.



## 3.4 Model validation and sensitivity analysis

To validate the model, we compared monthly mean values of riparian ET simulated with PERSiST with those obtained empirically from daily stream flow variations. Daily variations of stream flow can be used as a proxy for ET from near-stream zones (Cadol et al., 2012; Gribovszki et al., 2010) and they correlate well with direct sap flow measurements at the study site (Lupon et al., 2016). Daily stream flow variations measured at one particular point integrates riparian ET upstream from that point. Thus, we assumed that differences in specific daily stream flow variations between the up- and midstream sites, and the mid- and downstream sites were comparable to the specific riparian ET simulated with PERSiST for the midstream and downstream sub-catchments, respectively.

Once the best parameter set was obtained for the model configuration that included the riparian compartment, we further investigated the importance of riparian ET on regulating stream flow at Font del Regàs by using a Monte Carlo (MC) sensitivity analysis approach. This approach consisted in comparing sets of model efficiencies obtained from two MC analyses. In the first one, all model parameters potentially influencing stream flow were allowed to vary $\pm$ 25% with respect to the best performing parameter set from manual calibration (non-fixed ET analysis, Table S1). In the second one, landscape specific ET-related parameters (i.e. degree day rates, threshold temperatures, and ET adjustments) were kept constant, while the other parameters were again allowed to vary $\pm$ 25% (fixed ET analysis, Table S1). Fixed ET-related parameters were set to the mean optimal values obtained for each landscape unit after the manual calibration. The MC analyses consisted of 100 iterations of 1000 runs each. We used Tukey's Honestly Significant Difference tests (Tukey HSD) to compare log(NS) obtained from the 100 best runs (each from each of the iterations) between the fixed and non-fixed ET analyses. A decrease in the goodness of fit (i.e. lower values of log(NS)) for the fixed ET analysis was interpreted as an indication that the outputs of the model were sensitive to riparian ET, and thus, that the riparian component was important for simulating stream flow dynamics. The comparison between fixed and non-fixed ET analyses was made for the downstream site only, first for the overall calibration period, and then for the vegetative and dormant periods separately.



### 3.5 Modelling future projections of water budgets

The best manual parameterization of the model configuration including the riparian compartment was used to simulate future changes in catchment water budgets and to explore the contribution of riparian ET to these changes. We calculated future water balances considering predicted changes in climate for 2081–2100. Daily meteorological data for the period 1933–2000 was available from Turó de l'Home meteorological station, located < 10 km from Font del Regàs meteorological station (Meteocat,

www.meteocat.cat). Although Turó de l'Home is usually colder and wetter than Font del Regàs, monthly precipitation and temperature showed a strong correlation between the two stations for the period 2010–2014 (in the two cases: $R^2$ > 0.90, p < 0.001, n > 53, Figure S1). The linear regression models for these two sites were used to construct daily time series of temperature and precipitation at Font del Regàs for both the reference period (1981–2000) and the future period (2081–2100) based on Representative

Concentration Pathway (RCP) projections.

RCP projections provided by IPCC (2013) are based on the reference period 1986-2005. We assumed similar projections values for our reference period (1981–2000), which was the one for which data at Turó de l'Home was available. We applied the 2.5, 4.5, 6.0, and 8.5 RCP scenarios for Mediterranean zones including percentiles 0.25, 0.50, and 0.75 (IPCC, 2013). In general, RCP scenarios forecast an increase

in temperature all year round, but more pronounced in summer than in winter. Precipitation is predicted to decrease in April–September, while small changes are expected in October–March (Table 2).

For each year and RCP scenario, we calculated (i) the Aridity Index (AI) as a proxy of water availability (UNEP, 1992), and (ii) the relative contribution of simulated riparian ET to annual water catchment depletions at the whole catchment scale (i.e. the sum of simulated upland ET, riparian ET, and stream

flow at the downstream site). The AI is the relationship between annual precipitation and potential ET (PET), which was estimated using the Penman-Monteith equation on daily time steps (Allen et al., 1998). We assumed constant wind velocity (1 m s$^{-1}$) and relative humidity (75%) in the equation because these data were not recorded at the Turó de l'Home meteorological station. These assumptions were based on a 5-year time series of meteorological data at the Font del Regàs catchment (period 2010–2014; wind

velocity = 1.0 ± 0.4 m s$^{-1}$; relative humidity = 75 ± 9%). We examined the relationship between the



relative contribution of riparian ET to annual water catchment depletions and AI by fitting a two segment piecewise linear regression model. All the statistical analyses were carried out with the R 3.3.0 statistical software (R Core Team, 2012).

# 4 Results

## 4.1 Data–model fusion

For the calibration period (2010–2012), mean annual flow was $23 \pm 17$, $82 \pm 66$, and $105 \pm 113$ L s$^{-1}$ at the up-, mid-, and downstream sites, respectively. The three sites showed the same seasonal pattern, characterized by a strong decline in stream flow during the vegetative period (Figure 2). The model configuration excluding the riparian compartment successfully reproduced the magnitude and seasonal pattern of stream flow at the three sampling sites (Table 3 and Figure 2). However, there were mismatches between simulated and observed values, especially during the vegetative period, when stream flows were overestimated at the three sampling sites (RVD < 0, Table 3). The mismatches were especially noticeable in the downstream site, where simulated values were, on average, 53% higher than observed ones (Table 3).

The efficiency indexes indicated that the inclusion of the riparian compartment was essential to improve the agreement between simulated and observed flows at the mid- and downstream sites. The model including the riparian compartment showed higher NS and log(NS) metric values and RDV values closer to 0 (more accurate stream water volumes) than the one without riparian compartment (Table 3). Moreover, the model including the riparian compartment captured both the magnitude and seasonal pattern exhibited by stream flow even during low flow periods (Figure 2). On average, the inclusion of the riparian compartment reduced daily stream flow by 27% during the vegetative period at the downstream site (Table 3).

## 4.2 Model validation and sensitivity analysis

There was a good agreement between simulated daily rates of riparian ET and those obtained independently of model outputs for both the mid- and downstream sub-catchments. Simulated rates of





riparian ET were lower during the dormant ($0.89 \pm 0.97$ mm d$^{-1}$) than during the vegetative period ($3.7 \pm$ 1.3 mm d$^{-1}$). The lowest simulated ET values occurred in January and February (0.1–0.3 mm d$^{-1}$), while June and August showed the highest ones (5–7 mm d$^{-1}$) (Figure S2). The daily variation of stream flow followed a seasonal pattern similar to that exhibited by simulated daily riparian ET. Consequently, there was a strong and positive relationship between monthly mean values of simulated daily riparian ET and measured daily stream flow variations for both the midstream sub-catchment (linear regression [l.r.], $R^2$ = 0.83, p < 0.001, n = 24) and the downstream sub-catchment (l.r., $R^2$ = 0.88, p < 0.001, n = 24) (Figure 3).

The sensitivity analysis showed no differences in log(NS) values between the analysis with fixed and non-fixed ET parameters for the whole calibration period (Figure 4). The same occurred when comparing fixed and non-fixed ET simulations for the dormant period. For the vegetative period, the simulation of stream flow worsen when the ET parameters were fixed as indicated by the decrease in log(NS) efficiencies (Figure 4). Similar results were obtained for the NS metric (not shown).

**4.3 Present and future contribution of riparian ET to catchment water budgets**

Simulated rates of riparian ET averaged 931 mm yr$^{-1}$ for the calibration period (2010–2012) and contributed 5.91% to annual water losses. This contribution falls within the range of simulated values (5.54–8.42%) obtained for the reference period (1981–2000; mean annual riparian ET = $862 \pm 105$ mm). During both calibration and reference periods, the contribution of riparian ET to water catchment depletion was maximal from July to September, when it accounted for 8–19% of water catchment losses.

According to our simulations, mean annual riparian ET will increase in the future by 2% (scenario RCP 2.5 percentile 0.25) to 13% (scenario RCP 4.5 percentile 0.75), reaching mean annual rates of riparian ET between 879 and 977 mm yr$^{-1}$. This will represent an increase in the contribution of riparian ET to catchment water budgets of 1–2% compared to the reference period (Table 4). Moreover, future increases in warming and drying will smooth the seasonality of riparian ET by lengthening the vegetative period (ET rates > 0 mm d$^{-1}$) by 6 to 106 days (depending on the scenario and year) (Figure 5).



In the most moderate scenario (RCP 2.5 percentile 0.5), mean daily riparian ET values increased by 0.3 ± 0.1 mm d$^{-1}$ during the dormant period, which represents an increase of 19 ± 7 % compared to the reference period. During the vegetative period, the projected changes in mean daily riparian ET were smaller (-0.1 ± 0.1 mm d$^{-1}$) and represent a small fraction compared to the reference period (-2 ± 4 %) (Figure 5a and 5b). The most extreme scenario (RCP 8.5, percentile 0.5) simulated high riparian ET rates (> 2 mm d$^{-1}$) during most of the year. For this scenario, riparian ET rates increased by 0.6 ± 0.1 mm d$^{-1}$ during the dormant period, which represents an increase of 46 ± 16 % compared to the reference period. During the vegetative period, riparian ET rates decreased by -0.4 ± 0.6 mm d$^{-1}$. This is a decrease of 11 ± 22 % compared to the reference period (Figure 5g and 5h).

The AI decreased from 0.65 ± 0.18 to 0.45 ± 0.15 between the reference and the most extreme climate scenario (RCP 8.5, percentile 0.75). The contribution of riparian ET to catchment water budgets was low (6.40 ± 0.35 %) and unrelated to AI for AI > 0.83. Below this threshold, the contribution of riparian ET to catchment water budgets increased linearly up to 9.78% with decreasing AI. This dual behavior was well captured by a two segment linear regression relating AI and riparian ET contribution to catchment water depletion with a break point at AI = 0.83 ($R^2$ = 0.77, p < 0.001, n = 260) (Figure 6).

## 5 Discussion

### 5.1 Influence of riparian ET on stream flow and catchment water budgets

This study shows that the riparian zone contributed to regulate water exports and budgets at the Font del Regàs catchment. The inclusion of the riparian compartment in the PERSiST model structure improved the simulations, especially at the downstream site, where the riparian zone occupies 10% of the catchment area. These results support the idea that riparian zones are especially important on shaping stream flow dynamics at the valley bottom of mountainous catchments, likely due to the combination of a lower catchment connectivity (i.e. lower water inputs from uplands) (Bernal et al., 2012; Covino and McGlynn, 2007) and a greater water demand by riparian trees (Lupon et al., 2016).

In agreement with our expectations, the influence of the riparian zone on stream flow dynamics varied between seasons. During the dormant period, the model efficiencies barely improved when the riparian



compartment was included in PERSiST ($< 5\%$), suggesting a minimal effect of the riparian zone on stream flow dynamics (Table 3). Conversely, the riparian zone played a key role during the vegetative period because it contributed to drying up the stream as shown by the notable improvement in the log(NS) index,

which is a proxy of the goodness of fit during low flow conditions. Moreover, mismatches between simulated and observed water volumes decreased substantially (by 26%) if riparian ET was included, suggesting that the riparian zone can be important to regulate low flows in this catchment (Table 3). Although the inclusion of the riparian compartment contributed to significantly improve the goodness of fit, the model was not able to capture the lowest flows at the end of the vegetative period. Hydrological

processes not included so far in the PERSiST structure, such as the uptake of water by trees directly from the stream (Gribovszki et al., 2010; Tabacchi et al., 2000) or the reverse flux of water from the stream towards the riparian zone (Butturini et al., 2003; Rassam et al., 2006), could contribute to drop down stream flow at Font del Regàs, and therefore to the mismatches between observed and simulated flows. These hydrological processes have been shown to be relevant for reproducing stream flow dynamics in

Mediterranean and semiarid areas (e.g. Medici et al. 2008), and thus PERSiST could improve its ability to simulate stream flows in water limited catchments if these processes would be implemented in the model structure.

The sensitivity analysis indicated that riparian ET was an important hydrological process driving stream flow during the vegetative period because the model efficiency (log(NS) index) was significantly higher

when riparian ET parameters were allowed to vary. Moreover, our results suggest that the hydrological processes occurring in the riparian compartment could reduce daily stream flow by 48% during the vegetative period. This value is consistent with previous studies showing that riparian ET can reduce the amount of water entering to streams by 30–100% (Dahm et al., 2002; Folch and Ferrer, 2015; Kellogg et al., 2008; Lupon et al., 2016). Previous models have suggested that the transpiration process from

saturated riparian zones is essential to successfully represent the annual water balance of water-limited catchments (Medici et al., 2008; Tsang et al., 2014). On an annual basis, our simulations indicate that riparian ET can account for ~ 7% of annual catchment depletions at Font del Regàs (Table 4). The contribution of riparian ET to water budgets was especially noticeable during the dry period of the year, when it contributed as much as 19% to daily catchment depletions. These values are similar to those





estimated for other catchments with AI = 0.6–0.8 (Folch and Ferrer, 2015; Tsang et al., 2014; Wine and
Zou, 2012; Yeh and Famiglietti, 2008) and suggest that computations of catchment water budgets
neglecting riparian ET will overestimate catchment water resources.

Overall, PERSiST was able to successfully simulate stream flow dynamics in the studied Mediterranean
catchment, regardless of whether the model structure included or not the riparian compartment (log(NS)
> 0.81, RDV < 0.11). Moreover, the validation analysis supported the simulation results because the
model was able to successfully capture both the magnitude and the temporal patterns of riparian water
demand estimated with an independent empirical approach (Figure 3). The successful simulations
obtained for Font del Regàs provide evidence that PERSiST can be an appropriate tool for understanding
catchment hydrology as well as for exploring how specific hydrological processes, such as riparian ET,
can influence stream hydrology under different climatic conditions and future scenarios.

## 5.2 Future changes in riparian ET

Our simulations suggest that changes in climate projected for later in this century will influence both the
magnitude and temporal pattern of riparian ET rates in Font del Regàs. Riparian ET rates will decrease in
June–September and increase in November–May. Simulated decreases in riparian ET during the
vegetative period were related to lower soil water availability as a consequence of lower precipitation in
summer. In concordance, other studies in water-limited regions have shown that low ET rates in summer
could result from the disconnection between the water table and the active root zone depth (Baird and
Maddock, 2005; Serrat-Capdevila et al., 2007), which can accelerate leaf litter fall (Rood et al., 2008;
Sabater and Bernal, 2011) and promote stream desiccation (Medici et al., 2008; Serrat-Capdevila et al.,
2007).

On the other hand, the overall warmer temperatures predicted for winter months explain the projected
increase of riparian ET during this period. According to our simulations, the length of the vegetative
period could increase by 6 to 106 days depending on the applied scenario, mostly as a consequence of an
earlier onset of the leaf out period. The enlargement simulated by PERSiST is consistent with
observations of spring advancement reported in forests worldwide (e.g. Peñuelas and Boada, 2003;



Richardson et al., 2006) and strongly supports the idea that climate change has a marked effect on tree phenology. Our results also suggest that the effect of lengthening of the vegetative period will overwhelm the reduction of ET rates during summer, and that this change in seasonality will increase annual riparian water use by 2–13%. This warming-induced pattern is concordant with that reported for water-limited

riparian forests in southern USA (Bunk, 2012; Serrat-Capdevila et al., 2011).

Finally, we found that increases in annual riparian ET under a warmer climate may have a small effect (1–2%) on the relative contribution of riparian ET to annual catchment water budgets. The small effect predicted by the model was likely because warming also induced higher ET from upland forests (4 ± 11%). However, our hydrological model does not account for changes in vegetation community induced

by warming, a phenomenon that is expected to occur in areas experiencing increases in water stress (Benito-Garzón et al., 2008; García-Arias et al., 2014; Peñuelas and Boada, 2003, Walther et al., 2002). If water becomes limiting, especially in the upland environments, species capable to better adjust their evapotranspirative demand may be favored and become dominant (Engelbrecht et al., 2007), which would lead to decreases in ET from uplands compared to riparian zones. In fact, previous studies suggest that

the contribution of riparian ET to catchment water depletion can increase disproportionally with water limitation, and that a threshold exists at intermediate arid positions (i.e. AI = 0.8) (Lupon et al., 2016). Below this threshold, the contribution of riparian ET to water budgets can markedly increase up to 40%, though riparian zones usually occupy less than 10% of the total catchment area (Tockner and Stanford, 2002). Our simulations are in line with this idea and further suggest that riparian forests could switch

from energy-limited to water-limited systems as warming and drying increases in the future (Budyko, 1974; Creed et al., 2014).

## 6    Conclusions and Implications

This study highlights that riparian ET can influence stream flow dynamics and water budgets in Mediterranean catchments. We showed that including the riparian zone within the PERSiST model

configuration led to improved model efficiencies and more accurate simulation of stream flow dynamics, especially during summer. On an annual basis, riparian ET contributed by 7% to water catchment depletions, its contribution being especially noticeable (8–19%) during dry summer months. Our results



add to the growing body of knowledge showing that riparian hydrology is essential for understanding and predicting stream flow dynamics in catchments experiencing some degree of water limitation. Moreover, our climate simulations indicated that riparian ET could play a major role on catchment water budgets as water scarcity increases in the future. At Font del Regàs, for instance, projected decreases of annual stream flow by the end of this century (3–13%) could be accompanied by increases in riparian ET of the same order (2–13%). Similar predictions have been made for other water-limited catchments of America and Europe (Christensen et al., 2004; Rood et al., 2008; Serrat-Capdevila et al., 2007), forewarning the potential increase of ecological issues related to water scarcity in regions that are already water limited. Overall, this study highlights that the ecohydrology of riparian zones needs to be considered for a responsible management and conservation of water resources in Mediterranean catchments.

## Acknowledgments

We are thankful to Sílvia Poblador, Andrew Wade, and Martin Erlandsson for their invaluable field and modelling assistance. Special thanks are extended to Martyn Futter for his inspirational contributions. Financial support was provided by the Spanish Government through the projects MONTES-Consolider (CSD2008-00040-MONTES), MEDFORESTREAM (CGL2011-30590), and MEDSOUL (CGL2014-59977-C3-2). AL was supported by a Kempe Foundation stipend (Sweden). JLJL was funded by the Swedish Research Council (Svenska Forskningsrådet Formas, Grant/Award Number: 2015-1518). SB work was funded by the Spanish Research Council (JAE-DOC027), the Spanish CICT (Juan de la Cierva contract JCI-2008-177), European Social Funds (FSE), and the NICUS (CGL-2014-55234-JIN) project.

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



## Tables and Figures

**Table 1** Catchment drainage area, percentage of evergreen oak, decidious beech and riparian forest area, width of the riparian zone, and total basal area of riparian trees for the three nested sub-catchments considered in this study.

| | Sub-catchment characteristics | | | Riparian zone characteristics | |
| --- | --- | --- | --- | --- | --- |
| | Local drainage area (km²) | Evergreen (%) | Decidious (%) | Riparian (%) | Mean Width (m) | Total Basal Area (m² BA) |
| **Upstream** | 1.80 | 8.2 | 92.8 | 0.0 | -- | -- |
| **Midstream** | 6.74 | 52.6 | 42.5 | 4.9 | 12 | 822 |
| **Downstream** | 4.42 | 57.8 | 32.3 | 9.9 | 19 | 1354 |



**Table 2** Representative Concentration Pathway (RCP) projections for Mediterranean zones for 2081–
2100 as compared with the reference period 1981–2000. RCP values are indicated for each season for
temperature and for each semester for precipitation. Values are medians and interquartile ranges [25th,
75th percentiles] (IPCC, 2013).

| Projection | Temperature (ºC) | | | | Precipitation (%) | |
|---|---|---|---|---|---|---|
| | Dec–Feb | Mar–May | June–Aug | Sep–Nov | Oct–Mar | Apr–Sep |
| **RCP 2.5** | +1.25 [+0.75, +1.25] | +0.75 [+0.75, +1.25] | +1.25 [+0.75, +1.75] | +1.25 [+0.75, +1.75] | 0 [0, +5] | 0 [-5, 0] |
| **RCP 4.5** | +1.75 [+1.25, +2.50] | +1.75 [+1.25, +2.50] | +2.50 [+1.75, +3.5] | +2.50 [+1.75, +2.50] | 0 [-5, +5] | 0 [-15, 0] |
| **RCP 6.0** | +1.75 [+1.75, +2.50] | +2.50 [+1.75, +2.50] | +3.50 [+2.50, +4.50] | +2.50[+2.50, +3.50] | -5 [-15, 0] | -5 [-15, 0] |
| **RCP 8.5** | +3.50 [+2.50, +4.50] | +3.50 [+3.50, +4.50] | +6.00 [+4.50, +6.00] | +4.50 [+3.50, +6.00] | -5 [-15, 0] | -25 [-35, -15] |





**Table 3** Comparison between model calibrations including and excluding the riparian compartment. Log transformed Nash-Sutcliffe (NS) model efficiency coefficient and relative volume differences (RDV) of observed versus simulated stream flow (in parenthesis) at the up-, mid-, and downstream sites for vegetative, dormant, and whole calibration periods (September 2010–August 2012). Negative RDV values indicate an overestimation of modelled flow volumes compared to observed flow volumes, while positive RDV values indicate the opposite. The NS model efficiency values are not shown because they were similar to log(NS) values.

| | Vegetative | | Dormant | | All data | |
|---|---|---|---|---|---|---|
| | **No Riparian** | **Riparian** | **No Riparian** | **Riparian** | **No Riparian** | **Riparian** |
| **Upstream** | 0.56 (-0.19) | 0.56 (-0.19) | 0.82 (0.16) | 0.82 (0.16) | 0.82 (0.01) | 0.82 (0.01) |
| **Midstream** | 0.56 (-0.20) | 0.70 (-0.07) | 0.87 (0.15) | 0.89 (0.12) | 0.85 (0.09) | 0.89 (0.04) |
| **Downstream** | 0.00 (-0.53) | 0.49 (-0.27) | 0.90 (0.12) | 0.91 (0.07) | 0.81 (-0.11) | 0.88 (-0.05) |



**Table 4** Aridity index, annual riparian evapotranspiration (ET) rates, and relative contribution of riparian ET to annual catchment water depletions (i.e., upland ET + riparian ET + stream flow) for the reference period (1981–2000) and for each Representative Concentration Pathway (RPC) scenario during the future

period (2081–2100). Values are mean ± standard deviation.

| Scenario | Percentile | Aridity Index | Annual Riparian ET (mm) | Riparian ET Contribution (%) |
|---|---|---|---|---|
| **Reference** | | 0.65 ± 0.19 | 862 ± 105 | 7.09 ± 0.89 |
| **RCP 2.5** | 0.25 | 0.62 ± 0.20 | 879 ± 115 | 7.36 ± 0.93 |
| | 0.50 | 0.63 ± 0.20 | 910 ± 116 | 7.42 ± 0.94 |
| | 0.75 | 0.64 ± 0.20 | 936 ± 124 | 7.42 ± 0.93 |
| **RCP 4.5** | 0.25 | 0.59 ± 0.16 | 848 ± 120 | 7.67 ± 0.98 |
| | 0.50 | 0.60 ± 0.19 | 922 ± 128 | 7.68 ± 0.96 |
| | 0.75 | 0.62 ± 0.20 | 977 ± 136 | 7.68 ± 0.94 |
| **RCP 6.0** | 0.25 | 0.52 ± 0.14 | 826 ± 117 | 7.96 ± 0.96 |
| | 0.50 | 0.58 ± 0.16 | 934 ± 126 | 7.78 ± 0.93 |
| | 0.75 | 0.56 ± 0.18 | 969 ± 135 | 7.82 ± 0.93 |
| **RCP 8.5** | 0.25 | 0.50 ± 0.17 | 759 ± 132 | 8.25 ± 0.96 |
| | 0.50 | 0.53 ± 0.18 | 862 ± 145 | 8.16 ± 0.95 |
| | 0.75 | 0.45 ± 0.15 | 952 ± 160 | 8.22 ± 0.91 |





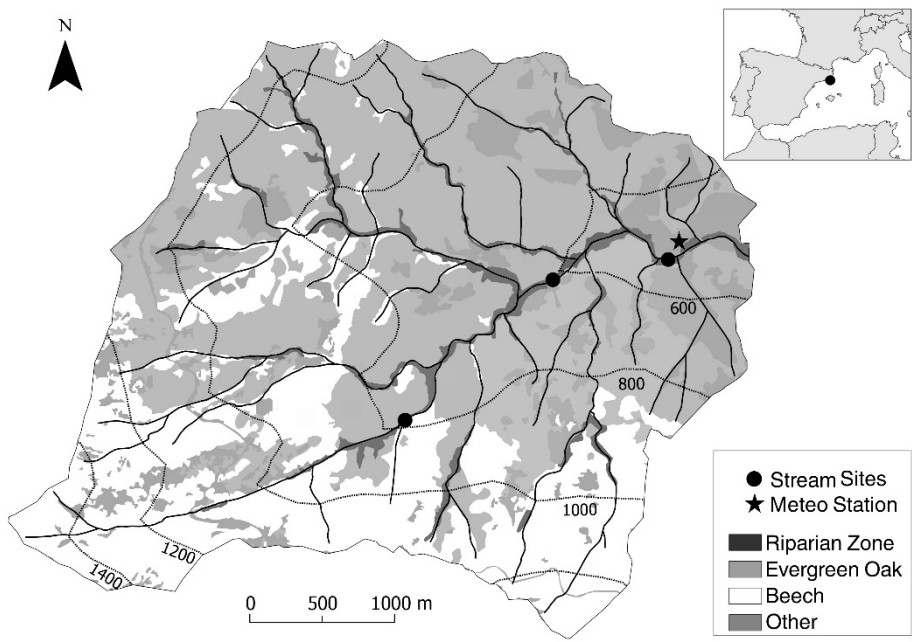

**Figure 1** Map of the Font del Regàs catchment (Montseny Natural Park, NE Spain). The location of the three nested stream sites (black circles) and the meteorological station where precipitation and temperature was measured (star) are shown.




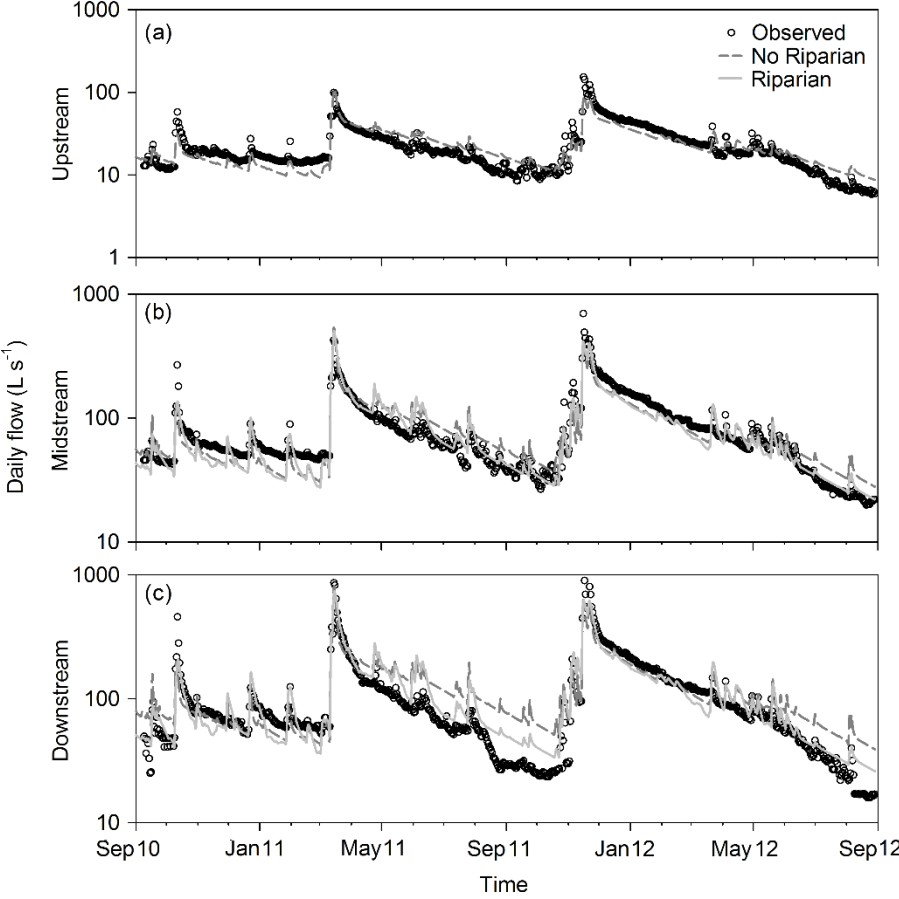

**Figure 2** Temporal pattern of stream flow for the (a) upstream, (b) midstream, and (c) downstream sites during the study period. Open circles represent observed values, while lines are simulated values excluding (dashed) and including (solid) the riparian compartment in the model configuration. Note that the upstream sub-catchment had no riparian forest, and therefore, simulations with and without riparian zone are equal.



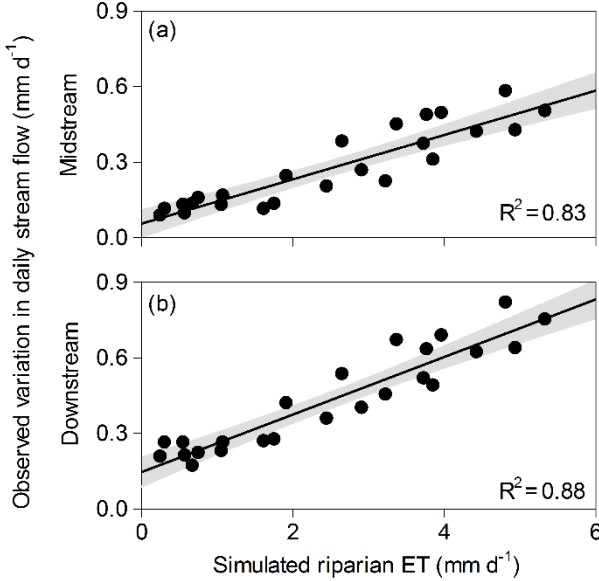

**Figure 3** Relationship between monthly mean values of simulated daily riparian evapotranspiration (ET) and observed daily stream flow variations (used here as an independent proxy of riparian ET) for (a) the midstream and (b) the downstream sub-catchments for the calibration period (September 2010–August 2012). The linear regression and the 95% confidence interval are also shown. For both mid- and downstream sites: p-value < 0.001, n = 24. The upstream sub-catchment had no riparian forest and it is not shown.





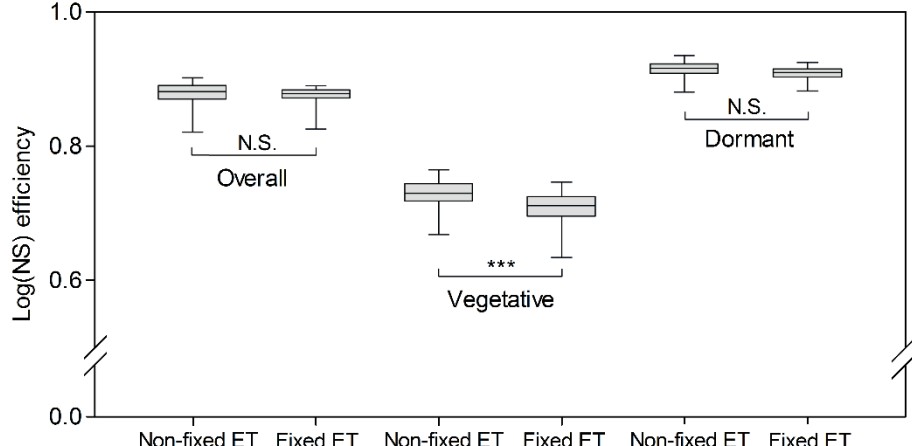

**Figure 4** Box plot of the 100 best log(NS) efficiencies obtained with the Monte Carlo (MC) simulations using the model configuration that included the riparian compartment at the downstream sub-catchment. MC analyses were performed using all potentially sensitive parameters first (Non-fixed ET), and fixing evapotranspiration-related parameters second (Fixed ET). Means of corresponding distribution pairs were compared using Tukey's Honestly Significant Difference tests. N.S. indicate no significant difference and *** indicate statistically significant difference (p<0.0001).





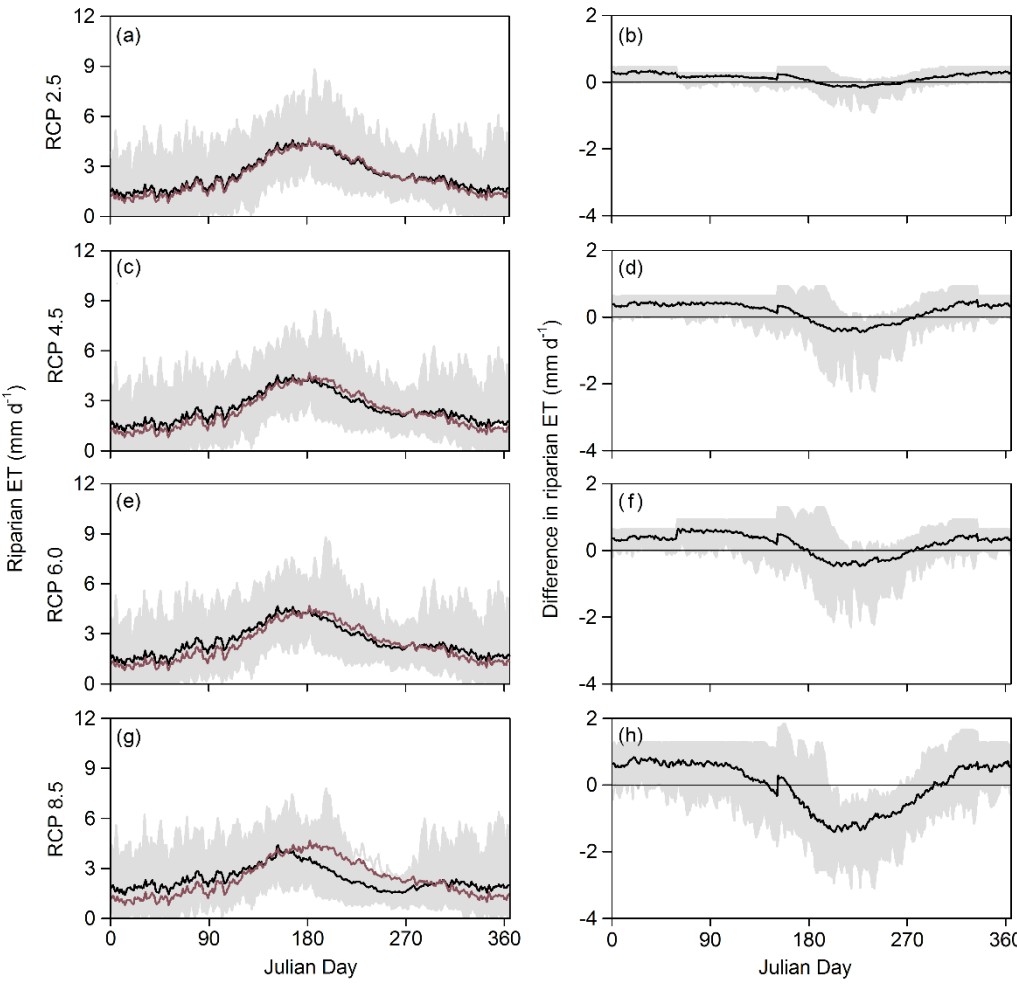

**Figure 5** Seasonal pattern of (left panels) daily riparian evapotranspiration rates simulated for different climate change scenarios and (right panels) difference in the simulated values of daily riparian evapotranspiration between the reference period (1981–2000) and future climate scenarios (2081–2100). All the climate change scenarios were based on the 0.5 percentile of the Representative Concentration Pathway (RCP) projections provided by IPCC (2013) for the period 2081–2100 (Table 2): (a,b) RCP 2.5 (the most moderate scenario), (c,d) RCP 4.5, (e,f) RCP 6.0, and (g,h) RCP 8.5 (the most extreme scenario). Black lines are mean values and grey shadows indicate the maximum–minimum range of values simulated for the 20-years period. The red line in the left panels is the mean daily values of riparian ET for the reference period. The horizontal line in the right panel is shown as a reference.





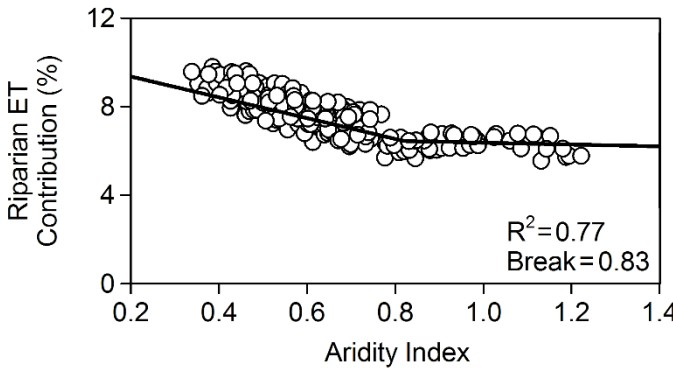

**Figure 6** Relationship between the relative contribution of riparian evapotranspiration (ET) to annual catchment water depletions and the aridity index for all the projections simulated with PERSiST as well as for the reference period. Total water output fluxes from the catchment (water depletions) are the sum of stream flow, upland ET, and riparian ET. The aridity index is the ratio between annual precipitation and potential evapotranspiration (P/PET). The goodness of fit of the two segment linear model and the break point are also shown.