# Peer review of "Title: Riparian evapotranspiration is essential to simulate stream flow dynamics and water budgets in a Mediterranean catchment"

_Hydrology and Earth System Sciences, 2017_

## Referee Comment (RC1) · Anonymous Referee #1 · 1 Feb 2018

The paper investigates the effect of riparian zones on hydrometric streamflow responses and catchment water budgets with a particular focus on riparian evapotranspiration. The authors use a semi-distributed conceptual bucket-type model to simulate a Mediterranean catchment with different setups. First, they demonstrate that the inclusion of a riparian compartment improves the model performance, especially during the vegetation period. Second, they demonstrate that the catchment response is sensitive to the evapotranspiration parameters of the riparian zone during the vegetation period. Third, they performed several climate scenario simulations to discuss the effect of riparian evapotranspiration on water budgets with climate change. Overall, the article is well structured, the text reads fluently and figures and tables are clear. I read

the paper with great interest. It nicely demonstrates that riparian zones and their ET should be considered in catchment models and I think studies like this are necessary to raise the hydrological model community's awareness for the role of riparian zones in a catchment. However, while reading I came across two major issues that concerned me several times throughout the text. These two major concerns and several minor issues should be addressed and clarified before publication.

Major issues:

1) The first issue is related to the aim of the study and the chosen approaches to accomplish it. In the introduction it is stated that it is known from several studies that riparian ET has in impact on stream flow dynamics and water budgets, but that there is a lack of respective studies at catchment scale. This suggests that the study focusses on the aspect of the catchment scale (such as the seasonal influence of riparian ET on hydrological connectivity between uplands and stream networks (cf. l.77-78) or the discussed percentage contribution of riparian ET to total catchment water depletion). Yet, large parts of the paper analyze and discuss the impact of riparian ET on stream flow dynamics without a clear relation to catchment scale specific aspects. Model validation follows the unusual idea of validating the performance of the riparian ET over the same period that was calibrated against discharge (and also some ET characteristics), instead of validating the performance of the calibrated response (discharge) for another period than the calibration period. I think this approach is valid since the performance of riparian ET is of specific interest for this study. Certainly, a validation of the discharge response would be good as well, especially since the model is used for climate scenario simulations where it is of interest that discharge (and ET) simulates well also under different conditions than experienced in the calibration period. However, my bigger concern is that model validation relies on the idea that daily variations of stream flow can be used as proxy for riparian ET. If the relation between riparian ET and streamflow dynamics is already approved enough to be used for the creation of validation data, this necessarily raises the question why the effect of riparian ET on

streamflow dynamics has to be analyzed in additional studies. Again, the introduction states that this effect is known, but title and large parts of the paper (partly even the introduction, cf. l. 71) read as if this is one of the main points of the study. Especially in the discussion section the results are mainly compared to agreeing studies of riparian ET and I missed a clear delineation in which way this study brings up new insight in the role of riparian ET for catchment water budgets and streamflow responses. In addition, the authors often use the inclusion/exclusion of the riparian compartment as equivalent to an inclusion/exclusion of riparian ET (l.22-23, l.143-145, l.158-160, l.326). In my opinion, the inclusion of the riparian compartment can only be used to analyze the effect of the riparian zone as a total, since the riparian compartment represents more fluxes than only ET. It is true that the model mainly improved during the vegetation period and that this suggests a major influence of riparian ET. However, at least the RDV improved also during the dormant season, which could be explained by the additional storage/buffer component of the riparian compartment. Moreover, a different parameterization of the riparian ET (less strong riparian ET compared to upland ET during the vegetation period) might have a different effect (e.g. similar improvement of the model during vegetative and dormant period). My suggestion would be to keep the presented methods and results unchanged, but to shift the focus in the discussion and introduction (and other explanations throughout the text) from the role of riparian ET on discharge dynamics to 1) the role of riparian zones and its ET for hydrological modelling of catchments and 2) how this might vary under different climate conditions.

2) The second main issue concerns the model setup. I especially had problems to understand how the three subcatchments were defined. According to the namings of the subcatchments (e.g. downstream subcatchment, downstream site), Table 1 and the way how validation data were calculated (l.197-201), I understood the subcatchments as three individual parts summing up to the total catchment. According to the description of the calibration data (l.134-140), the aim of the study (influence of riparian ET in a catchment) and some applied methods and presented results, I guess the subcatchments include the total upstream drainage area (i.e. the downstream subcatchment is equivalent to the total catchment). Besides a clarification of the definition of the subcatchments in the text, I think a figure showing the conceptual setup of the models would be very useful. Such a figure would also make it easier to understand the differentiation between landscape units, layers and compartments and the flux connections between them (especially for l.145-160). Additionally, I missed a more detailed description of the model parameters and the represented fluxes. Since the study focusses on the influence of ET, at least the conceptualisation of ET and the related ET parameters (degree day rates, threshold temperature parameters) should be explained in more detail in the text and / or in a figure. For example, it is discussed that the length of the vegetative period increased in the climate scenarios at that this was mostly a consequence of a changed tree phenology, i.e. an earlier onset of the leaf out period, thus tree phenology (l.371-380). It is not clear to me if and how the length of the vegetation period and the tree phenology (e.g. leaf out period) were considered in the model structure and thus it is difficult to follow the argumentation.

Minor comments:

3) I suggest to change the title to: How riparian evapotranspiration shapes stream flow dynamics and water budgets in a Mediterranean catchment model, cf. comment 1)

4) l.25: Shouldn't it be the same value as in l.286?

5) l.28-29: I would consider more relevant that this increases the contribution of riparian ET to catchment water depletion by 1-2%

6) l.36-37, l.47-48: Please provide some references

7) L46-47: Why only in regions potentially suffering from water scarcity? An explanation is coming in l.58-59, maybe this can be put closer together (e.g. moving l.44-48 at the end of the second paragraph). A small rearrangement of the two first paragraphs of the introduction could also prevent that the sentence in l.49-50 seems somehow contradictory to the first part of the introduction (l. 36-39).

8) l.76-78: If I understood the functioning of the used model correctly, the connectivity between uplands and stream networks is mainly controlled by the riparian zone and its ET. In that case, the model setup (higher riparian ET during the vegetation period) makes this expectation somehow self-evident.

9) Figure 1: The color code in the legend (riparian zone = black) does not match the colors in the map (riparian zone = dark grey)

10) l.91: Upland means only the part covered by beech forests and heathlands or all the catchment except of the riparian zone? Please clarify

11) l.94: Increases 12-fold compared to what?

12) l.92 and l.97: Are there also B and C horizons?

13) l.98-107: This describes the subcatchments clearly as three independent subcatchments. If it is meant in a different way (cf. comment 2), please clarify in this section

14) l.114: 'other catchment water pools' is identical to landscape units? Or to soil layers? Or to the upland compartment? And which are the water fluxes represented in these other water pools, also subsurface flow and ET?

15) L.122-123 'a specified fraction of rainfall can be directly transported to stream runoff': Does this mean overland flow? Or is it direct precipitation on the stream? If it is the latter, shouldn't it also be accounted for during wet conditions?

16) l.152-157: From the description I understand that overland flow was basically disabled. Why is it then necessary to include a layer representing overland flow (l.149)?

17) l.176: I would expect different values for the riparian ET-related parameters than for the upland ET-related parameters in order to allow different ETs. However, in Table S1 the best riparian and upland ET-related parameters seem to be identical

18) l.184-185: Do you refer to all water fluxes other than ET? In addition, it is very

difficult to make use of the given information about the adapted parameters, without a more detailed description (cf. comment 2)

19) Both in section 3.4 and 3.5 it is not clear which of the 3 model instances including a riparian compartment are used. I assume it is the downstream sub-catchment (in the sense of being the total catchment, cf. comment 2), but please specify

20) l.202: This sounds like you refer to model calibration, however, it is confusing since you talk about validation in the paragraph above

21) l.210: Why the ET parameters are fixed to mean values of the landscape units instead of taking the optimal parameter for each landscape unit?

22) l.211-213: I do not understand the formulation '100 iterations of 1000 runs'. Does it mean you tested 100 times 1000 different parameter sets? If yes, what was the criteria to split the total of 100000 simulations in sets of 1000?

23) l.214-216: I think it is difficult to restrict this effect to riparian ET. It should be related to ET in general, both from the upland and riparian compartments, since the ET parameters were fixed for both compartments

24) l.253: I would be careful to say that the strong decline in stream flow is characteristic for the vegetative period only. In 2012 the stream flow is declining from the beginning of the year. Maybe it would be good to include the precipitation time series in Figure 2 in order to explain this behavior

25) l.256-257: Complementary there were underestimations at all three sampling sites for the dormant season, which were in similar RDV ranges for the up- and midstream catchment but much lower in the downstream catchment compared to the vegetative period. It would be great to mention and discuss this, also with regard to the improvements that were achieved for the vegetative and dormant season with the inclusion of the riparian compartment (cf. comment 1 and 32).

26) Figure 2 would be clearer with reduced sizes of the observation points

27) l.265: Please specify which the low flow periods are. This will also help to distinguish between the low flow periods (captured) and the lowest flows (not captured) (l. 329).

28) l.266: In Table 3, l.23 and l.326 you give a value of 26%. Even though, I am not sure that this is a correct formulation. It should be 'reduced daily stream flow by 26 percentage points' or 'reduced stream flow overestimations to 27 % during the vegetative period'. See also l. 340-342, where you give a different percentage value, which is actually the correct one when talking about a change in percentage compared to RDV = -0.53 as reference.

29) l.293: Also here I guess it should 1-2 percentage points?

30) l.294-295: Is your definition of the vegetative period really an ET rate > 0 mm/d ? During the dormant season there should normally also be days with ET > 0 mm/d. Moreover, for the model performance calculations you define the vegetative period as ranging from April-October (l.192).

31) Figure 5 and l.296-304: What about the 0.25 percentile and 0.75 percentile scenarios? Shouldn't the RCP 2.5 percentile 0.25 be the most moderate and the RCP 8.5 percentile 0.75 be the most extreme scenario?

32) l.321-322: For log(NS) I agree, for RDV I would say there was an improvement also during the dormant season. This could be related to riparian effects (fluxes and additional storage) other than ET (cf. also comment 1 and 25) and should be discussed.

33) l.340 'when riparian ET parameters were allowed to vary': Also the uphill ET parameters were allowed to vary or fixed (cf. comment 23). It should be discussed, why this setup allows to conclude on the riparian ET only.

34) L353-355: This sounds like if it is superfluous to consider the riparian compartment

35) l.405: You show that there is an effect of riparian ET on the catchment water budget (8-19%) and that this effect can slightly increase (1-2%), but I would not say that you

can call this a major control (cf. also your discussion l.381-389)

36) l.406-407: Maybe I missed it, but I cannot remember that you mentioned this before

---

## Referee Comment (RC2) · Anonymous Referee #2 · 19 Feb 2018

General comments: This paper seeks to determine the influence of riparian evapo-transpiration (ET) in streamflow dynamics and the prediction of water budgets at a catchment scale. The authors used a flexible landscape scale rainfall-runoff model to simulate daily stream exports with and without the influence of riparian ET. The results demonstrate that when the riparian ET compartment is considered in the model, then the prediction across seasons and sub-catchments are improved. Moreover, the article studies the influence of this compartment under climate scenarios and demonstrates that riparian ET could play a significant role when estimating catchment hydrology with respect to climate change, especially under extreme drought conditions. The paper is well-written and straightforward, and the scientific findings represent a valuable contribution to the field. I recommend publication with minor revisions.

Specific comments:

My main concern in this manuscript relates to the model's description. With the information provided, it is difficult to follow how each piece of information falls into place for the simulation. For example, L135-138 mention procedures which include water pressure sensors to determine stream water level, the use of an ISCO sampler, and an empirical relationship between flow and water level using a slug chloride addition. The connection between these sentences seems unclear; was the water level data measured while conducting tracer injections? It is also not clear to me if these parameters were used as model inputs or if they were used further to compare between the observed and the simulated streamflows (L144). If they do not belong to the model inputs, I recommend placing that information in a different section. I believe section 3.3, "Calibration procedure," could start at L141 (remove redundant information from L162), and then the technical information regarding streamflow evaluation could be included in L144. If, on the other hand, my interpretation about the use of these parameters is inaccurate and they belong to section 3.2, please clarify their role as model inputs and include how the observed streamflow data was gathered (e.g. nearby gaging station, instream discharge measurements).

In addition, it seems unclear if the model is capable of considering different types of vegetation and its influence on riparian ET. It would be very useful to condense the information expressed in sections 3.2 and 3.3. I also recommend the creation of a conceptual figure or table that lists all the variables used for input, calibration, and the model output, as well as a short description of ET related parameters and model capabilities in term of predictions regarding vegetation changes. Consider replacing the titles within the "Materials and methods" section to: 3.2 Model inputs and 3.3 Model configuration and calibration procedure.

Technical corrections:

[Figure]

L48: Please provide some references.

Figure 1: The location map on the top right needs more context; it would be useful to label key landmarks (i.e. names of countries or cities) for better reference. The color code used is difficult to follow. It is hard to identify the areas where the riparian zone is present since the color selected is masked by the color used for stream delineation (in the printed version, the riparian zone color code looks black). It might also be useful to number your stream sites in the figure and then add the corresponding label in the legend (e.g. 1 Upstream; 2 Midstream; 3 Downstream). The map also includes contour lines that seem to be representative of the catchment elevation, however, these are not mentioned in the figure caption, please clarify.

L147: Consider changing "divided" to "categorized".

L162: Insert "in the literature" after "ET values reported".

L165: Change "the" to "model" in "Note that the instances...".

L175-176: It is unclear how the authors defined dormant and vegetative period. Please add more clarification in regards to this. Also, L192 attributes the specific month of the data set to the periods under discussion. It would be more useful to state this classification the first time the periods were mentioned in the text (i.e. L175-176).

L195: Is riparian ET one of PERSiST's outputs, or was it calculated using modeled streamflow data? So far, only streamflow (catchment water fluxes) has been introduced as a model result. Please briefly list output parameters of interest under the model description in section 3.1.

L294: Is the length of vegetative period determined only by the simulated values of ET (e.g. ET rates > 0 mm d-1)? I think this could be clearer with more insights on what the authors used to classify this period.

L373-374: It is unclear how the extension of the vegetative period in the climate model's scenarios can be associated to early onset of the leaf out period after considering the

limitations of the model in L330-334. Please clarify.

L376: The role of vegetation in the model's performance or predictive capabilities has been understated throughout the text, hence arguing that model results "strongly support" an effect of climate change in tree phenology seems uncertain. Please provide clarification or references that help support this statement.

L384-385: This seems to contradict the statement on L376.

---

## Referee Comment (RC3) · Anonymous Referee #3 · 19 Feb 2018

The paper by Lupon, et al. uses a hydrological runoff model to examine the importance of evapotranspiration in riparian zones on water budgets in several catchments. The description of the exercise was well written and generally easy to follow, although the agonizing detail (necessary, but no less agonizing) of the model testing and calibration makes this paper quite a chore to work through. Given that demonstrating that the model does a good job of predicting flow in the catchments studied is certainly important, it may be difficult to cut the highly detailed exposition. In the end, however, that detail overshadows the actual results obtained when the model was exercised to address the question. I would like to see the authors place more emphasis on the outcome of the exercise so as to help readers who may not need the detailed methods to

find and appreciate what the authors have generated. Indeed, some of the modelling detail might be placed into supplementary material.

The paper makes a very useful statement, but there are supporting reports of empirical work that the authors could use to support the conclusions of their work in the absence of original data. In particular, a paper by Flewelling et al. (Hydrol. Proc., 2013, doi:10.1002/hyp.9763) shows exactly what the effect of near-field evapotranspiration can have on water delivery to the adjacent stream, and to biogeochemical reactions occurring in the stream sediments. It is entirely consistent with the present manuscript.

The use of the Nash Sutcliffe Index is appropriate here, but many people will not recognize it. Because this paper should have a broad audience, the N-S index should be defined better. Give the equation – I had to look it up, as it was new to me.

Other reviewers have provided a detailed, line by line commentary on the manuscript. Given my general agreement with those comments, I will not repeat them here.

---

## Author Comment (AC1) · 27 Mar 2018

Dear editor and reviewers of *Hydrology and Earth System Sciences*,

Please find enclosed our responses to the reviewers' comments regarding the paper "Riparian evapotranspiration shapes stream flow dynamics and water budgets in a Mediterranean catchment" (hess-2017-735), which in the new version will be entitled "Riparian evapotranspiration is essential to simulate stream flow dynamics and water budgets in a Mediterranean catchment". Overall, we feel happy with the positive reviews and that the reviewers found the study interesting and a potential contribution to *Hydrology and Earth System Sciences* journal. Their suggestions as well as their editing corrections have been of great help for us.

We have thoughtfully read all their comments and we are working on the manuscript to incorporate them all. Following the reviewers' suggestion, we have carefully rewritten both the introduction and discussion sections to better frame our work within the modelling community as well as to better highlight the novelties of the study. Moreover, we do now better describe the data used to calibrate the model, provide a detail definition of the vegetative period, and explain how the three sub-catchment were characterized. We have also included new supplementary materials with detailed information of the model conceptualization (including a new conceptual figure) and parametrization in order to better describe the model set up and, at the same time, clarify the model description in the main text. Finally, we do now explicitly address the influence of the riparian zone on flow simulations during the dormant period in both the results and discussion sections.

Below you will find detailed responses to each of your general comments as well as to the most substantial specific comments. Overall, we believe that we can successfully solve all the points raised by the reviewers, and generate an improved version of the manuscript including all their suggestions.

Please, do not hesitate to contact us if further clarifications are needed at this stage.

Sincerely,

Anna Lupon

CC: José L. J. Ledesma and Susana Bernal

**Anonymous Referee #1**

**General comment:** *The paper investigates the effect of riparian zones on hydrometric streamflow responses and catchment water budgets with a particular focus on riparian evapotranspiration. The authors use a semi-distributed conceptual bucket-type model to simulate a Mediterranean catchment with different setups. First, they demonstrate that the inclusion of a riparian compartment improves the model performance, especially during the vegetation period. Second, they demonstrate that the catchment response is sensitive to the evapotranspiration parameters of the riparian zone during the vegetation period. Third, they performed several climate scenario simulations to discuss the effect of riparian evapotranspiration on water budgets with climate change. Overall, the article is well structured, the text reads fluently and figures and tables are clear. I read the paper with great interest. It nicely demonstrates that riparian zones and their ET should be considered in catchment models and I think studies like this are necessary to raise the hydrological model community's awareness for the role of riparian zones in a catchment. However, while reading I came across two major issues that concerned me several times throughout the text. These two major concerns and several minor issues should be addressed and clarified before publication.*

**Answer:** Many thanks for your positive comments. We are glad that you enjoyed the paper and that you consider that "studies like this are necessary" to improve catchment hydrological models. We deeply appreciate your tremendously detailed and constructive review. We have carefully considered all your suggestions and worked to incorporate them in the new version of the manuscript.

**Major issues:**

1) *The first issue is related to the aim of the study and the chosen approaches to accomplish it. In the introduction it is stated that it is known from several studies that riparian ET has an impact on stream flow dynamics and water budgets, but that there is a lack of respective studies at catchment scale. This suggests that the study focusses on the aspect of the catchment scale (such as the seasonal influence of riparian ET on hydrological connectivity between uplands and stream networks (cf. L77-78) or the discussed percentage contribution of riparian ET to total catchment water depletion). Yet, large parts of the paper analyze and discuss the impact of riparian ET on stream flow dynamics without a clear relation to catchment scale specific aspects. Model validation follows the unusual idea of validating the performance of the riparian ET over the same period that was calibrated against discharge (and also some ET characteristics), instead of validating the performance of the calibrated response (discharge) for another period than the calibration period. I think this approach is valid since the performance of riparian ET is of specific interest for this study. Certainly, a validation of the discharge response would be good as well, especially since the model is used for climate scenario simulations where it is of interest that discharge (and ET) simulates well also under different conditions than experienced in the calibration period. However, my bigger concern is that model validation relies on the idea that daily variations of stream flow can be used as proxy for riparian ET. If the relation between riparian ET and streamflow*

*dynamics is already approved enough to be used for the creation of validation data, this necessarily raises the question why the effect of riparian ET on streamflow dynamics has to be analyzed in additional studies. Again, the introduction states that this effect is known, but title and large parts of the paper (partly even the introduction, cf. L71) read as if this is one of the main points of the study. Especially in the discussion section the results are mainly compared to agreeing studies of riparian ET and I missed a clear delineation in which way this study brings up new insight in the role of riparian ET for catchment water budgets and streamflow responses. In addition, the authors often use the inclusion/exclusion of the riparian compartment as equivalent to an inclusion/exclusion of riparian ET (L22-23, L143-145, L158-160, L326). In my opinion, the inclusion of the riparian compartment can only be used to analyze the effect of the riparian zone as a total, since the riparian compartment represents more fluxes than only ET. It is true that the model mainly improved during the vegetation period and that this suggests a major influence of riparian ET. However, at least the RDV improved also during the dormant season, which could be explained by the additional storage/buffer component of the riparian compartment. Moreover, a different parameterization of the riparian ET (less strong riparian ET compared to upland ET during the vegetation period) might have a different effect (e.g. similar improvement of the model during vegetative and dormant period). My suggestion would be to keep the presented methods and results unchanged, but to shift the focus in the discussion and introduction (and other explanations throughout the text) from the role of riparian ET on discharge dynamics to 1) the role of riparian zones and its ET for hydrological modelling of catchments and 2) how this might vary under different climate conditions.*

**Answer:** We thank the reviewer for highlighting this issue, and also for suggesting how to improve the introduction and discussion, which has been very helpful. As the reviewer pointed, previous studies have already shown that processes occurring in riparian zones can drive diel and seasonal patterns in stream flow (e.g. Flewelling et al., 2014; Lupon et al., 2016; Rassam et al., 2006). However, there are few hydrological catchment models explicitly considering the riparian compartment, which ultimately limits our ability to quantify the influence of riparian zones on stream flow and catchment water export across regions. Specifically, applying hydrological models that consider the riparian compartment to water limited catchments could be a helpful tool to better understand how riparian zones can shape catchment water budgets and availability for both in- and off-stream uses, as well as to achieve feasible predictions of hydrological and ecological responses to future climate. In this sense, our study demonstrates that the riparian zone is a key compartment to properly simulate both catchment hydrology and stream flow dynamics, and consequently, that this landscape unit should be considered in hydrological models. Moreover, the successful simulations obtained for Font del Regàs provide evidence that hydrological models can be an appropriate tool for exploring how specific hydrological processes, such as riparian ET, can influence stream hydrology under different climatic conditions. Following the reviewer suggestion, we have rewritten parts of the introduction and discussion sections to better highlight this point. Moreover, we now state that the aim of our study was to explore the role of riparian ET on successfully simulating present and future stream flow dynamics and catchment water exports in a Mediterranean forested headwater catchment. With these improvements, we believe that both introduction and discussion are now better

framed within the context of hydrological modelling and explicitly address the rational that was implicit in our study design.

We also agree with the reviewer that the inclusion/exclusion of the riparian compartment is not equivalent to an inclusion/exclusion of riparian ET because there are other riparian processes (e.g. longer water travel times) that can additionally affect stream flow responses. We carefully checked all the manuscript to avoid confusions. Moreover, we do now highlight that, during the dormant period, the model efficiency (RDV) improved from +0.12 to +0.07 when the riparian zone was included. This result suggests that increased water storage within riparian zones can also influence stream flow dynamics during wet conditions.

Finally, we discarded the idea to split the time series in order to calibrate and validate against two independent data sets. First, and similar to other authors (Oreskes et al., 1994), we are skeptical about the possibility of meaningful validation of environmental models, especially when the model performs better for the validation data set than for the calibration data set. Moreover, our data set was relatively short and showed strong inter- and intra-annual variability, which make difficult to split the data in order to have a full range of environmental conditions for both the calibration and validation procedures. Thus, we are more confident to get a more robust parametrization for both present and future projections by using the entire available dataset for model calibration. This procedure has been shown to be appropriate and successful in previous studies (e.g., Larssen et al. 2007; Ledesma and Futter, 2017). We chose to use daily variations in stream flow to validate the simulated riparian ET rates, which is one of the main outputs of the model together with stream flow. These daily variations in stream flow were independent from the model input data and further, they are robust estimates of riparian ET (e.g. Flewelling et al., 2014; Lupon et al. 2016). Thus, we believe they were a neat, alternative way to validate model results. Following your suggestion, we have clarified the model validation procedure in the text.

2) *The second main issue concerns the model setup. I especially had problems to understand how the three sub-catchments were defined. According to the naming of the sub-catchments (e.g. downstream sub-catchment, downstream site), Table 1 and the way how validation data were calculated (L197-201), I understood the sub-catchments as three individual parts summing up to the total catchment. According to the description of the calibration data (L134-140), the aim of the study (influence of riparian ET in a catchment) and some applied methods and presented results, I guess the sub-catchments include the total upstream drainage area (i.e. the downstream sub-catchment is equivalent to the total catchment). Besides a clarification of the definition of the sub-catchments in the text, I think a figure showing the conceptual setup of the models would be very useful. Such a figure would also make it easier to understand the differentiation between landscape units, layers and compartments and the flux connections between them (especially for L145-160). Additionally, I missed a more detailed description of the model parameters and the represented fluxes. Since the study focusses on the influence of ET, at least the conceptualization of ET and the related ET parameters (degree day rates, threshold temperature parameters) should be explained in more detail in the text and/or in a figure. For example, it is discussed that the length of the vegetative period increased in the climate scenarios at that this was mostly a*

*consequence of a changed tree phenology, i.e. an earlier onset of the leaf out period, thus tree phenology (l.371-380). It is not clear to me if and how the length of the vegetation period and the tree phenology (e.g. leaf out period) were considered in the model structure and thus it is difficult to follow the argumentation.*

**Answer:** We agree with the reviewer that further clarification was needed in this regard. The three sub-catchments are nested, and thus, the downstream sub-catchment is equivalent to the total drainage area. Given that the flow at each sampling site integrates all processes occurring within its drainage area, the PERSiST model simulate flows based on (i) the flow coming from the upstream nested-catchment and (ii) the proportion of each landscape unit of the local drainage area. For example, flows at the downstream sub-catchment outlet were simulated based on midstream sub-catchment flows and the proportion of evergreen, deciduous, and riparian forests in the local drainage area of the downstream sub-catchment. To avoid confusions, we have clarified this issue throughout the manuscript. Now, we refer to (i) local vs. total drainage area and (ii) single nested sub-catchment vs. whole catchment.

We have produced new supplementary information to clarify the model set up and conceptualization. This includes tables for model inputs, model calibration data, relevant model parameters (and corresponding description), and model outputs. Moreover, and following the reviewers' suggestion, we have included a conceptual figure illustrating the model structure including and excluding the riparian compartment. Finally, we have clarified and better structured the material and methods subsections related to model description, configuration, and calibration. Specifically, we do now explain that PERSiST conceptualizes the landscape in four spatial levels: catchment, sub-catchment, landscape unit, and bucket/soil box.

Finally, we agree with both reviewers #1 and #2 that the association between the extension of the vegetative period and riparian tree phenology was not clearly explained. Simulations for future climate change scenarios showed that the number of days with ET > 0 mm d$^{-1}$ would mostly increase during spring. The reason for such increase is a larger amount of days with temperatures above the "growing degree threshold", a parameter indicating the temperature threshold above which ET occurs. We have rewrite the discussion in order to clarify this issue. Moreover, we do now state that these results suggest a potential enlargement of the vegetative period, which is consistent with previous observations showing that climate change can affect riparian tree phenology by promoting the advancement of the riparian leaf out period (Perry et al., 2012; Serrat-Capdevila et al., 2007).

**Minor comments:**

*3) I suggest to change the title to: How riparian evapotranspiration shapes stream flow dynamics and water budgets in a Mediterranean catchment model, cf. comment (1)* **Answer:** Thanks for the suggestion; we have changed the title to "Riparian evapotranspiration is essential to simulate stream flow dynamics and water budgets in a Mediterranean catchment".

4) L25: *Shouldn't it be the same value as in L286?* **Answer:** Right, thanks for noticing. We changed the value to "5.5–8.4%".

5) L28-29: *I would consider more relevant that this increases the contribution of riparian ET to catchment water depletion by 1-2%.* **Answer:** Ok, we do now state that "Simulations considering climate change scenarios suggest that the contribution of riparian ET to annual water budgets would increase from 7.1% to 8.2% as water scarcity increases in the future".

6) L36-37, l.47-48: Please provide some references **Answer:** Ok (e.g. Kampf and Burges, 2007; Ledesma and Futter, 2017).

7) L46-47: *Why only in regions potentially suffering from water scarcity? An explanation is coming in L58-59, maybe this can be put closer together (e.g. moving L44-48 at the end of the second paragraph). A small rearrangement of the two first paragraphs of the introduction could also prevent that the sentence in L49-50 seems somehow contradictory to the first part of the introduction (L36-39).* **Answer:** Following the reviewer suggestion, we moved this sentence at the end of the second paragraph. The first paragraph now points out that despite previous field-based studies have shown that riparian zones can affect seasonal and diel variations in stream flow (e.g. Flewelling et al., 2014; Lupon et al., 2016; Rassam et al., 2006), there are still few hydrological catchment models explicitly considering the riparian compartment, which ultimately limits our ability to quantify the influence of riparian zones on stream flow and catchment water export across regions.

8) L76-78: *If I understood the functioning of the used model correctly, the connectivity between uplands and stream networks is mainly controlled by the riparian zone and its ET. In that case, the model setup (higher riparian ET during the vegetation period) makes this expectation somehow self-evident.* **Answer:** That's right, the expectation was quite obvious. We have removed it.

9) *Figure 1: The color code in the legend (riparian zone = black) does not match the colors in the map (riparian zone = dark grey).* **Answer:** Right, we changed the color of the riparian zone in the legend. Moreover, and following reviewer #2 suggestions, we changed the color code of the figure, labeled the countries, and numbered the stream sites. We hope that those changes improved the visualization and conceptualization of the figure.

10) L91: *Upland means only the part covered by beech forests and heathlands or all the catchment except of the riparian zone? Please clarify.* **Answer:** Upland represent all land covers except of the riparian zone (i.e., evergreen oak and beech forests). We do now clarify the "upland" meaning in the text.

11) L94: *Increases 12-fold compared to what?* **Answer:** We meant that total basal area of riparian trees increases by 12-fold from headwaters to the valley bottom. We modified the text to clarify that both riparian width and total basal area of riparian trees markedly increase along the catchment.

12) L92 and L97: *Are there also B and C horizons?* **Answer:** We analyzed the soil profile of the top 120-150 cm of the soil in both upland and riparian forests. For these range of depths, we identified horizons O, A, and B. This information has been included to the study site section.

13) L98-107: *This describes the sub-catchments clearly as three independent sub-catchments. If it is meant in a different way (cf. comment 2), please clarify in this section.* **Answer:** They were three nested sub-catchments. We clarified this issue along the text (please, see our answer to your comment #2 for more information).

14) L114: *'other catchment water pools' is identical to landscape units? Or to soil layers? Or to the upland compartment? And which are the water fluxes represented in these other water pools, also subsurface flow and ET?* **Answer:** "Catchment water pools" referred to catchment compartments (e.g. upland forest, riparian zone, and stream). We have clarified this point in the text. Moreover, we have built a new figure in the supplementary material that conceptualizes the model set up.

15) L.122-123 *'a specified fraction of rainfall can be directly transported to stream runoff': Does this mean overland flow? Or is it direct precipitation on the stream? If it is the latter, shouldn't it also be accounted for during wet conditions?* **Answer:** We meant that, under very dry conditions, a fraction of rainfall can be directly transported to stream runoff via overland flow. We have rewritten the sentence to clarify this point.

16) L152-157: *From the description I understand that overland flow was basically disabled. Why is it then necessary to include a layer representing overland flow (L149)?* **Answer:** Good point. By design, PERSiST always needs to include a "quick bucket" that receives water from precipitation. From this layer, water can be transported to the stream (i.e., overland flow) and/or percolated to the upper soil box. Based on previous knowledge from the field, we considered that all water percolated to the upper soil layer. We have clarified this issue in the manuscript.

17) L176: *I would expect different values for the riparian ET-related parameters than for the upland ET-related parameters in order to allow different ETs. However, in Table S1 the best riparian and upland ET-related parameters seem to be identical.* **Answer:** This is a good remark, neatly caught by the reviewer. Values shown in former Table S1 are actually correct (i.e., identical values for riparian and upland ET-related parameters). This is because, unfortunately, ET-related parameters are configured as landscape unit-specific, whereas riparian zone is configured as an extra "soil box" that communicates the upland soil compartment with either groundwater or stream compartments. Thus, in the model configuration including the riparian zone, each of the landscape units (evergreen or deciduous) had both an upland and a riparian box, but only one set of ET-related parameters associated to both boxes. To be able to simulate realistic ET values during parameterization for the three forest types (i.e, evergreen, deciduous, and riparian), we tuned two different soil box-specific parameters: (i) the "Retained water depth", a parameter representing the water depth in a soil box below which water no longer freely drains but can be lost via ET; and (ii) the "Time constant" parameter, which represents the water

residence time within a soil box. By giving higher values of those two parameters in the riparian soil box compared to the upland soil box, we could simulate higher ET rates in the riparian compartment as a result of (i) a greater water availability and (ii) a longer water residence times in the riparian than in the upland soil boxes. These two phenomenon likely occur in the reality as a result of changes in soil texture and the proximity to streams, and thus, we feel confident that it was an effective way to simulate different ET rates given the model constrains. New supplemental materials (Supplement 1 and 2) have been crafted in order to better describe the model configuration.

18) L184-185: *Do you refer to all water fluxes other than ET? In addition, it is very difficult to make use of the given information about the adapted parameters, without a more detailed description (cf. comment 2).* **Answer:** We adjusted all model parameters (including those related to ET) to optimize the overall fit between observed and simulated hydrographs. This point has been clarified in the text. Moreover, we do now include a set of supplementary information to better explain the model description (please, see our reply to comment #2).

19) *Both in section 3.4 and 3.5 it is not clear which of the 3 model instances including a riparian compartment are used. I assume it is the downstream sub-catchment (in the sense of being the total catchment, cf. comment 2), but please specify.* **Answer:** The sensitivity analyses (section 3.4) were performed only for the downstream site because it integrates the entire catchment dynamics. On the other hand, the contribution of riparian ET on total catchment depletions (section 3.5) was calculated at the whole catchment scale. To do so, we summed up values of simulated upland ET and riparian ET in appropriate proportions at the three sub-catchments, plus the stream flow at the downstream site (i.e. catchment outlet). We have rewritten the methods section to clarify which sub-catchment (and why) was considered in each case.

20) L202: *This sounds like you refer to model calibration, however, it is confusing since you talk about validation in the paragraph above.* **Answer:** As we previously mentioned (response to comment #1), we discarded the possibility to validate model performance by splitting the data set. Our time series was relatively short and span a large range of environmental conditions, and thus, we decided to use the whole data set for the calibration in order to obtain a more robust (and realistic) parameter set for future simulations. Please, see our response to you comment #1 for more information on this regard.

21) L210: *Why the ET parameters are fixed to mean values of the landscape units instead of taking the optimal parameter for each landscape unit?* **Answer:** We fixed the ET parameters to mean values in order to have a set of homogenous non-calibrated values that we can use as a "control" in the sensitivity analyses. If the optimal values would have been used, model performance would have not been independent of ET parameter values, as they would have already been tuned (calibrated) to maximize model performance.

22) L211-213: I *do not understand the formulation '100 iterations of 1000 runs'. Does it mean you tested 100 times 1000 different parameter sets? If yes, what was the criteria to split the total of 100000 simulations in sets*

*of 1000?* **Answer:** Yes, we tested 100 times, 1000 different parameter sets. We split the total number of simulations because the MC tool only retain the best parameter set (in terms of model efficiency provided) from each of the iterations for the further analysis (i.e., the sensitivity analysis). This has been clarified in the manuscript. Moreover, we chose to run 100 iterations because it has been shown that running more than 100 iterations does not add extra information.

23) L214-216: *I think it is difficult to restrict this effect to riparian ET. It should be related to ET in general, both from the upland and riparian compartments, since the ET parameters were fixed for both compartments.* **Answer:** That's right. We have clarified this point throughout the manuscript. Yet, we would like to highlight that the largest decrease in model performance for the fixed ET analysis was obtained for the riparian vegetative period. Thus, we believe that it is reasonable to relate the model sensitivity analysis with the role of riparian ET on model performance.

24) L253: *I would be careful to say that the strong decline in stream flow is characteristic for the vegetative period only. In 2012 the stream flow is declining from the beginning of the year. Maybe it would be good to include the precipitation time series in Figure 2 in order to explain this behavior.* **Answer:** That's right; we do now state that the seasonal pattern was characterized by lower stream flow during the vegetative than during the dormant period. Moreover, and following the reviewer suggestion, we have included the temporal pattern of precipitation in Figure 2.

25) L256-257: *Complementary there were underestimations at all three sampling sites for the dormant season, which were in similar RDV ranges for the up- and midstream catchment but much lower in the downstream catchment compared to the vegetative period. It would be great to mention and discuss this, also with regard to the improvements that were achieved for the vegetative and dormant season with the inclusion of the riparian compartment (cf. comment 1 and 32).* **Answer:** Following the reviewer suggestion, we do now state that "during the dormant period, the inclusion of the riparian compartment reduced the underestimation of stream flow from +12% to +7%" at the downstream site". Also, we do now discuss these results in section 5.1 (please see our responses to comment #1 and #32).

26) *Figure 2 would be clearer with reduced sizes of the observation points.* **Answer:** Ok.

27) L265: P*lease specify which the low flow periods are. This will also help to distinguish between the low flow periods (captured) and the lowest flows (not captured) (L329).* **Answer:** Ok (June-September).

28) L266: *In Table 3, L23 and L326 you give a value of 26%. Even though, I am not sure that this is a correct formulation. It should be 'reduced daily stream flow by 26 percentage points' or 'reduced stream flow overestimations to 27 % during the vegetative period'. See also L340-342, where you give a different percentage value, which is actually the correct one when talking about a change in percentage compared to RDV = -0.53 as reference.* **Answer:** Following the reviewer suggestion, we do now state that "the inclusion of the riparian

compartment reduced daily stream flow overestimations to 27% during the vegetative period at the downstream site". We have also clarified this result in the abstract.

29) L293: *Also here I guess it should 1-2 percentage points?* **Answer:** To avoid confusions, we do now state that "the contribution of riparian ET to catchment water budgets could increase from 7.1% (reference period) to 8.2% (scenario RCP 8.5 percentile 0.75)".

30) L294-295: *Is your definition of the vegetative period really an ET rate > 0 mm/d? During the dormant season there should normally also be days with ET > 0 mm/d. Moreover, for the model performance calculations you define the vegetative period as ranging from April-October (L192).* **Answer:** We agree with both reviewers #1 and #2 that the vegetative period could have been better defined. We consider that the vegetative period was comprised between the beginning of the riparian leaf-out (April) and peak of leaf litter fall (October), which coincided with the onset and offset of riparian tree transpiration, respectively. This definition can be found in the methods section of the new manuscript. Moreover, and in order to avoid confusions, we have rewritten the results to better explain that future increases in warming and drying will "smooth the seasonality of riparian ET and increase the number of days with ET rates > 0 mm d$^{-1}$".

31) *Figure 5 and l.296-304: What about the 0.25 percentile and 0.75 percentile scenarios? Shouldn't the RCP 2.5 percentile 0.25 be the most moderate and the RCP 8.5 percentile 0.75 be the most extreme scenario?* **Answer:** Following the reviewer advice, Figure 5 does now show the percentile 0.25 and 0.75 for RCP 2.5 and 8.5 scenarios, respectively.

32) L321-322: *For log(NS) I agree, for RDV I would say there was an improvement also during the dormant season. This could be related to riparian effects (fluxes and additional storage) other than ET (cf. also comment 1 and 25) and should be discussed.* **Answer:** Agreed. We do now argue that, during the dormant period, the inclusion of the riparian compartment improved the simulation of stream flow volumes. These results suggest that the riparian zone can be important for shaping stream flows during wet conditions, likely because it contributes to increase water storage, and thus water residence time, within the catchment.

33) L340 *'when riparian ET parameters were allowed to vary': Also the uphill ET parameters were allowed to vary or fixed (cf. comment 23). It should be discussed, why this setup allows to conclude on the riparian ET only.* **Answer:** That is correct, thanks. Please, see our response to comment #17 for a detailed explanation on this issue.

34) L353-355: *This sounds like if it is superfluous to consider the riparian compartment.* **Answer:** That's right; we removed this sentence from the manuscript. Thanks.

35) L405: *You show that there is an effect of riparian ET on the catchment water budget (8-19%) and that this effect can slightly increase (1-2%), but I would not say that you can call this a major control (cf. also your discussion l.381-389).* **Answer:** Ok, we have toned down our conclusion.

36) L406-407: *Maybe I missed it, but I cannot remember that you mentioned this before.* **Answer:** That's right. We do now mention in the results that "future climate change scenarios predict that upland ET would increase from 4% to 11% compared to the reference period, while stream flow would decrease from 3% to 13%".

**Anonymous Referee #2**

**General comment:** *This paper seeks to determine the influence of riparian evapotranspiration (ET) in streamflow dynamics and the prediction of water budgets at a catchment scale. The authors used a flexible landscape scale rainfall-runoff model to simulate daily stream exports with and without the influence of riparian ET. The results demonstrate that when the riparian ET compartment is considered in the model, then the prediction across seasons and sub-catchments are improved. Moreover, the article studies the influence of this compartment under climate scenarios and demonstrates that riparian ET could play a significant role when estimating catchment hydrology with respect to climate change, especially under extreme drought conditions. The paper is well-written and straightforward, and the scientific findings represent a valuable contribution to the field. I recommend publication with minor revisions.*

**Answer:** Thank you for your positive and constructive comments! We feel flattered that you find our study interesting.

**Specific comments:**

(1) *My main concern in this manuscript relates to the model's description. With the information provided, it is difficult to follow how each piece of information falls into place for the simulation. For example, L135-138 mention procedures which include water pressure sensors to determine stream water level, the use of an ISCO sampler, and an empirical relationship between flow and water level using a slug chloride addition. The connection between these sentences seems unclear; was the water level data measured while conducting tracer injections? It is also not clear to me if these parameters were used as model inputs or if they were used further to compare between the observed and the simulated stream flows (L144). If they do not belong to the model inputs, I recommend placing that information in a different section. I believe section 3.3, "Calibration procedure," could start at L141 (remove redundant information from L162), and then the technical information regarding streamflow evaluation could be included in L144. If, on the other hand, my interpretation about the use of these parameters is inaccurate and they belong to section 3.2, please clarify their role as model inputs and include how the observed streamflow data was gathered (e.g. nearby gaging station, instream discharge measurements).*

**Answer:** We agree that this part of the methods could be better explained. As the reviewer pointed, stream flow data was used to calibrate the model (not as model inputs). To clarify this issue, we changed the heading of section 2.3 to "model inputs, calibration data, and configuration". Moreover, we do now explicitly state that "we calibrated PERSiST to match stream flow data for two complete hydrological years at the outlet of the up-, mid-, and downstream sub-catchments". Finally, we better explain that stream flow was measured in situ with water pressure sensors (Teledyne Isco, Model 1612; more details in Lupon et al., 2016).

(2) *In addition, it seems unclear if the model is capable of considering different types of vegetation and its influence on riparian ET. It would be very useful to condense the information expressed in sections 3.2 and 3.3. I also recommend the creation of a conceptual figure or table that lists all the variables used for input, calibration, and the model output, as well as a short description of ET related parameters and model capabilities in term of predictions regarding vegetation changes. Consider replacing the titles within the "Materials and methods" section to: "3.2 Model inputs" and "3.3 Model configuration and calibration procedure".*

**Answer:** Following this suggestion as well as that from reviewer #1, we have now included substantial new materials that addresses the reviewer concerns (see our response to comment #2 of reviewer #1). To improve clarification, section 3.2 is now named "3.2 Model inputs, calibration data, and model configuration for the present study".

**Technical corrections:**

L48: *Please provide some references.* **Answer:** Ok, we included Flewelling et al. (2014); Lupon et al. (2016), and Rassam et al. (2006).

Figure 1: *The location map on the top right needs more context; it would be useful to label key landmarks (i.e. names of countries or cities) for better reference. The color code used is difficult to follow. It is hard to identify the areas where the riparian zone is present since the color selected is masked by the color used for stream delineation (in the printed version, the riparian zone color code looks black). It might also be useful to number your stream sites in the figure and then add the corresponding label in the legend (e.g. 1 Upstream; 2 Midstream; 3 Downstream). The map also includes contour lines that seem to be representative of the catchment elevation, however, these are not mentioned in the figure caption, please clarify.* **Answer:** Following the reviewer suggestion, we changed the color code of the figure, labeled the countries, and numbered the stream sites. Moreover, we do now specify in the caption that (i) dotted lines indicate the catchment elevation and (ii) the inset map shows the location of the Font del Regàs catchment within Spain. We believe that these changes will clarify the figure.

L147: *Consider changing "divided" to "categorized".* **Answer:** Ok.

L162: *Insert "in the literature" after "ET values reported".* **Answer:** Ok.

L165: *Change "the" to "model" in "Note that the instances...".* **Answer:** Ok.

L175-176: *It is unclear how the authors defined dormant and vegetative period. Please add more clarification in regards to this. Also, L192 attributes the specific month of the data set to the periods under discussion. It would be more useful to state this classification the first time the periods were mentioned in the text (i.e. L175-176).* **Answer:** We agree with both reviewers #1 and #2 regarding this point. Please, see our earlier response to reviewer #1 (comment #30).

L195: *Is riparian ET one of PERSiST's outputs, or was it calculated using modeled streamflow data? So far, only streamflow (catchment water fluxes) has been introduced as a model result. Please briefly list output parameters of interest under the model description in section 3.1.* **Answer:** Yes, riparian ET is one of the PERSIST's model outputs. The model provides daily values of ET or each soil box. PERSiST also provides simulated daily values of stream flow, water depth in soil boxes, and percolating water between soil boxes. Following your suggestion, we do now provide a brief list of output parameters in the supplementary information and refer to it in the manuscript if needed.

L294: *Is the length of vegetative period determined only by the simulated values of ET (e.g. ET rates > 0 mm d-1)? I think this could be clearer with more insights on what the authors used to classify this period.* **Answer:** No, the vegetative period was determined by the onset and offset of riparian tree transpiration (i.e. from the beginning of the riparian leaf-out to the peak of leaf litter fall). To avoid further confusions, we do now provide a definition of "vegetative period" in the method section.

L373-374: *It is unclear how the extension of the vegetative period in the climate model's scenarios can be associated to early onset of the leaf out period after considering the limitations of the model in L330-334. Please clarify.* **Answer:** We agree with both reviewers #1 and #2 in this regard. The simulations show that, in the future, the number of days with ET > 0 mm d$^{-1}$ will increase during spring as a consequence of warmer temperatures. Please, see the response to the comment #1 from reviewer #1 for further information.

L376: *The role of vegetation in the model's performance or predictive capabilities has been understated throughout the text, hence arguing that model results "strongly support" an effect of climate change in tree phenology seems uncertain. Please provide clarification or references that help support this statement.* **Answer:** Following the reviewer advice, we have toned down our statement. We do now argue that our results suggest a potential enlargement of the vegetative period, and that this idea is consistent with previous observations showing that climate change can affect riparian tree phenology by promoting the advancement of the riparian leaf out period (Perry et al., 2012; Serrat-Capdevila et al., 2007).

L384-385: *This seems to contradict the statement on L376.* **Answer:** That's right. Our model simulations showed that future warming will increase the number of days with riparian ET > 0 mm d$^{-1}$, which is consistent with previous observations showing a potential enlargement of the riparian vegetative period in the future (e.g.

Perry et al., 2012). However, our model was not able to simulate changes in vegetation community, a phenomenon that will likely occur in the studied region due to future drought conditions (e.g. Peñuelas and Boada, 2003). The enlargement of the vegetation period and the change in vegetation community are not exclusive, and thus, they can occur simultaneously in the future. We have clarified these two ideas in the main text.

**Anonymous Referee #3**

**General comment:** *The paper by Lupon, et al. uses a hydrological runoff model to examine the importance of evapotranspiration in riparian zones on water budgets in several catchments. The description of the exercise was well written and generally easy to follow, although the agonizing detail (necessary, but no less agonizing) of the model testing and calibration makes this paper quite a chore to work through. Given that demonstrating that the model does a good job of predicting flow in the catchments studied is certainly important, it may be difficult to cut the highly detailed exposition. In the end, however, that detail overshadows the actual results obtained when the model was exercised to address the question. I would like to see the authors place more emphasis on the outcome of the exercise so as to help readers who may not need the detailed methods to find and appreciate what the authors have generated. Indeed, some of the modelling detail might be placed into supplementary material.*

**Answer:** We are happy the reviewer considers our manuscript "well written and easy to follow" and has a general positive overview of it. Despite we acknowledge that the model description is quite long, we also agree with reviewers #1 and #2 that some extra information regarding the model configuration is needed to fully understand the results. Following their suggestion, we created new supplementary material that contained a detailed description of the model parameters as well as a new conceptual figure showing the model configuration. At the same time, we have clarified and better structured the model descriptions in the main text. We believe that with those changes, the model configuration will be easier to understand and, certainly, less agonizing.

**Specific comments:**

*(1) The paper makes a very useful statement, but there are supporting reports of empirical work that the authors could use to support the conclusions of their work in the absence of original data. In particular, a paper by Flewelling et al. (Hydrol. Proc., 2013, doi:10.1002/hyp.9763) shows exactly what the effect of near-field evapotranspiration can have on water delivery to the adjacent stream, and to biogeochemical reactions occurring in the stream sediments. It is entirely consistent with the present manuscript.* **Answer:** We do now include the Flewelling et al. (2014) paper to support our statements in both the introduction and discussion sections. Thanks for the suggestion!

*(2)   The use of the Nash Sutcliffe Index is appropriate here, but many people will not recognize it. Because this paper should have a broad audience, the N-S index should be defined better. Give the equation – I had to look it up, as it was new to me.* **Answer:** That's right, the Nash Sutcliffe Index is a common index used to evaluate hydrological models, but not all the audience should know it. We have clarified it in the text and we have reported some references that explain in detail the calculations (e.g., Nash and Sutcliffe, 1970). However, we decided to not include the formula in the main manuscript to avoid including more details in the methods section.

*(3)   Other reviewers have provided a detailed, line by line commentary on the manuscript. Given my general agreement with those comments, I will not repeat them here.* **Answer:** Ok, thanks.

**References**

Flewelling, S. A., Hornberger, G. M., Herman, J. S., Mills, A. L. and Robertson, W. M.: Diel patterns in coastal-stream nitrate concentrations linked to evapotranspiration in the riparian zone of a low-relief, agricultural catchment, Hydrol. Process., 28, 2150–2158, doi: 10.1002/hyp.9763, 2014.

Krampt, S. K. and Burges, S. J.: A framework for classifying and comparing distributed hillslope and catchment hydrologic models, Water Resour. Res., 43, W05423, doi:10.1029/2006WR005370, 2007.

Larssen, T., Høgåsen, T. and Cosby B. J.,: Impact of time series data on calibration and prediction uncertainty for a deterministic hydrogeochemical model, *Ecological Modelling*, *207*(1), 22-33, 2007

Ledesma, J. L. J., and Futter, M. N.: Gridded climate data products are an alternative to instrumental measurements as inputs to rainfall-runoff models, Hydrol. Process., 31, 3283-3293, 10.1002/hyp.11269, 2017.

Lupon, A., Bernal, S., Poblador, S., Martí, E. and Sabater, F.: The influence of riparian evapotranspiration on stream hydrology and nitrogen retention in a subhumid Mediterranean catchment, Hydrol. Earth Syst. Sci., 20, 3831–3842, doi:10.5194/hess-20-3831-2016, 2016.

Nash, J. E. and Sutcliffe, J. V.: River flow forecasting through conceptual models. Part 1: A discussion of principles, J. Hydrol., 10, 282–290, 1970.

Oni, S. K., Futter, M. N., Molot, L. A. and Dillon, P. J.: Adjacent catchments with similar patterns of land use and climate have markedly different dissolved organic carbon concentration and runoff dynamics, Hydrol. Process., 28, 1436-1449, 2014.

Oreskes, N., Shraderfrechette, K. and Belitz, K.: Verification, validation, and confirmation of numerical models in the earth sciences, *Science*, *263*(5147), 641-646, 1994.

Peñuelas, J. and Boada, M.: A global change-induced biome shift in the Montseny mountains (NE Spain), Glob. Chang. Biol., 9, 131–140, 2003.

Perry, L. G., Andersen, D. C., Reynolds, L. V., Nelson, S. M. and Shafroth, P. B.: Vulnerability of riparian ecosystems to elevated CO2 and climate change in arid and semiarid western North America, Glob. Chang. Biol., 18, 821–842, doi: 10.1111/j.1365-2486.2011.02588.x, 2012.

Rassam, D. W., Fellows, C. S., De Hayr, R., Hunter, H. and Bloesch, P.: The hydrology of riparian buffer zones; two case studies in an ephemeral and a perennial stream, J. Hydrol., 325, 308–324, doi:10.1016/j.jhydrol.2005.10.023, 2006.

Serrat-Capdevila, A., Valdés, J. B., González-Pérez, J., Baird, K., Mata, L. J. and Maddock, T.: Modeling climate change impacts – and uncertainty – on the hydrology of a riparian system: The San Pedro Basin (Arizona/Sonora), J. Hydrol., 347, 48–66, doi:10.1016/j.jhydrol.2007.08.028, 2007.

---

## Referee Report (RR1)

The authors have successfully revised the manuscript. Both the objective of the study and the model setup and functioning are much clearer now and all comments given by myself and the other two reviewers were addressed satisfactorily. I would recommend publication with some final minor technical corrections.

Technical corrections

L25-L26: The given numbers refer to the calibration plus reference period (cf. L294-297). Add this information in the sentence (e.g. '... of annual water depletions over a 20 years reference period (1981-2000) ...')

L26-L28: It should be clarified that the increase in the contributions to the water budgets is only small (cf. your statement in L.398-399). Maybe you could add a percentage (as done in the previous version) both here and in L398-399. In addition: 'the driest years' are never mentioned explicitly elsewhere in the manuscript, so either remove the phrase here or add the information in the results/discussion (e.g. Fig. 5).

L77-L79: I would suggest to mention here that you test the model with and without the riparian compartment

L92-L93: Rewrite to: Soils of heathlands, oak and beech forests are sandy with a 3 cm deep 0 horizon followed by a 5-15 cm deep A horizon and a > 100 cm deep B horizon. Rewrite L98-L99 accordingly.

L95: Remove the s of increases

L104 and L108: Replace at with in

L118/19, L123, Supplement1: Recheck the formulations. It is a bit confusing if the riparian forests are a landscape unit (see supplement) or a catchment compartment ( = ‚bucket').

L124-L125: This sentence is not very clear to me, try to rephrase it.

L128: Maybe use the same parameter names in Tab S3

L128-131: Consider to remove or reformulate the two sentences. In the present form I only understood their meaning with the help of Tab. S3

L135: Remove calibration data (since you also use ET values for soft calibration, just mentioned in 3.3). You could even move the first two sentences of this paragraph to the next section (to L167) and avoid to mention 'calibration' in this section.

L144: Supplement 2

L171-172: It would be great if you would mention which parameters for ET were adjusted here or if you at least mention it somewhere in Supplement S3 for Tab. S4 (cf. my former comment #17 and your reply to it).

L215: Substitute text with test

L227: Substitute 1933 with 1981 ?

L256: I think the precipitation data really help to understand the streamflow behaviour better (cf. former comment #24), showing that it is an interplay between precipitation input and the season (thus ET). Unfortunately I also wouldn't agree that the streamflow is lower during the vegetative

period than during the dormant period and I would suggest you simply remove the sentence L255-256.

L269: I would call 'low flow period' the period between August and October (cf. L345-346). Therefore I would suggest to replace 'even during low flow periods (June-September), especially in 2012' with 'except at the end of the vegetative period (August-October)'

L300: You have to recalculate the percentage value. The lowest value you obtain is 826 mm/yr, which is lower than the value for your reference period and thus not 2% (which corresponds to the value of 879mm/yr, which you indicated in the previous version manuscript). Also adapt it in L396 and L425.

L304: The value for scenario RCP 8.5 percentile 0.25 is higher (8.25%)

L306: It is not possible to see this increase of days with ET > 0 mm/d in Figure 5, since you only show the mean ET over the reference period (which never goes below 0 mm/d!) and not the ET for single years of the reference period. Either remove the cross-reference to Figure 5 or consider including some/all single years of the reference period in the figure.

L326-L327: This is the only sentence in the manuscript where it is still confusing what you considered as the downstream site (because the 10% riparian zone refer to the local drainage area of 4.42 km$^2$, the model output at the downstream site integrates the flow of all the drainage area). I would suggest to rephrase it.

L335: Add: at the downstream site

L365: I would suggest to remove 'saturated'

L373: components instead of component

L378: Maybe remove 'projected for later in this century'

L390: scenario and year (cf. L306)

L419-L420: You should give the same values here as in the abstract (L25-26); additionally consider to remove 'dry'

---

## Author Response (AR2)

Dear Dr. Nunzio Romano and Reviewer 1,

Many thanks for the additional technical notes on the manuscript entitled "Riparian evapotranspiration is essential to simulate stream flow dynamics and water budgets in a Mediterranean catchment" (hess-2017-735). We have considered them in the new revised version of the manuscript and the supplementary materials. Briefly, we have checked all the values along the manuscript to ensure that we provide the same results in the abstract, results, and discussion. Furthermore, we agree that some sentences needed further clarification, and we have rewritten them accordingly. Finally, your editing suggestions have been also included in the text.

We hope these changes correctly address your comments.

Sincerely,

Anna Lupon

CC: José L. J. Ledesma and Susana Bernal

**Reviewer 1:**

*The authors have successfully revised the manuscript. Both the objective of the study and the model setup and functioning are much clearer now and all comments given by myself and the other two reviewers were addressed satisfactorily. I would recommend publication with some final minor technical corrections.* **Answer:** Many thanks for the additional comments on the manuscript.

*L25-L26: The given numbers refer to the calibration plus reference period (cf. L294-297). Add this information in the sentence (e.g. "… of annual water depletions over a 20 years reference period (1981-2000) …").* **Answer:** Ok, we have changed the sentence accordingly (L25).

*L26-L28: It should be clarified that the increase in the contributions to the water budgets is only small (cf. your statement in L.398-399). Maybe you could add a percentage (as done in the previous version) both here and in L398-399. In addition: "the driest years" are never mentioned explicitly elsewhere in the manuscript, so either remove the phrase here or add the information in the results/discussion (e.g. Fig. 5).* **Answer:** We have clarified that "climate change scenarios suggest small increases in the contribution of riparian ET to annual water budgets" (L26-28). Moreover, we now specify the percentage of increase (i.e., 1–2%) both in the abstract and in the discussion sections (L28, L394). Finally, we have removed the sentence referring to the driest years.

*L77-L79: I would suggest to mention here that you test the model with and without the riparian compartment.* **Answer:** Ok (L82).

*L92-L93: Rewrite to: Soils of heathlands, oak and beech forests are sandy with a 3 cm deep 0 horizon followed by a 5-15 cm deep A horizon and a > 100 cm deep B horizon. Rewrite L98-L99 accordingly.* **Answer:** Ok (L95-96, L100-102).

*L95: Remove the s of increases.* **Answer:** Ok (L98).

*L104 and L108: Replace at with in.* **Answer:** Ok (L107).

*L118/19, L123, Supplement 1: Recheck the formulations. It is a bit confusing if the riparian forests are a landscape unit (see supplement) or a catchment compartment (= "bucket").* **Answer:** We have rewritten the supplements to clarify this issue (Supplement 1).

*L124-L125: This sentence is not very clear to me, try to rephrase it.* **Answer:** We have clarified that "from the upper soil box, water can infiltrate to lower soil boxes (e.g. groundwater), move laterally to the riparian zone or the stream, or return to the atmosphere via ET" (L125-126).

*L128: Maybe use the same parameter names in Tab S3* **Answer:** This sentence is no longer needed. Thanks anyway!

*L128-131: Consider to remove or reformulate the two sentences. In the present form I only understood their meaning with the help of Tab. S3* **Answer:** Ok, removed.

*L135: Remove calibration data (since you also use ET values for soft calibration, just mentioned in 3.3). You could even move the first two sentences of this paragraph to the next section (to L167) and avoid to mention "calibration" in this section.* **Answer:** We have removed "calibration data" from both the subheading and the sentence. However, we decided to keep these two sentences here to clarify the data used (as suggested by reviewer #2).

*L144: Supplement 2.* **Answer:** Ok, thanks (L143)

*L171-172: It would be great if you would mention which parameters for ET were adjusted here or if you at least mention it somewhere in Supplement S3 for Tab. S4 (cf. my former comment #17 and your reply to it).* **Answer:** For soft calibration, all parameters were slightly modified to simulate realistic values of evapotranspiration (ET). Yet, it is true that some parameters had a major effect (e.g. "degree day ET", "growing degree threshold", "ET adjustment" and "retained water depth"). We now mention these "key" parameters in the Supplements (Table S3).

*L215: Substitute text with test.* **Answer:** Ok.

*L227: Substitute 1933 with 1981?* **Answer:** Yes, 1981. Thanks for noticing (L225).

*L256: I think the precipitation data really help to understand the streamflow behaviour better (cf. former comment #24), showing that it is an interplay between precipitation input and the season (thus ET). Unfortunately I also wouldn't agree that the streamflow is lower during the vegetative period than during the dormant period and I would suggest you simply remove the sentence L255-256.* **Answer:** We agree with the reviewer that the seasonal pattern of stream flow is an interplay between precipitation and ET. Accordingly, we have modified the sentence as follow: "The three sites showed the same seasonal pattern, characterized by high flows during rain events and low flows in summer" (L253-254).

*L269: I would call "low flow period" the period between August and October (cf. L345-346). Therefore I would suggest to replace "even during low flow periods (June-September), especially 2012" with "except at the end of the vegetative period (August-October)"* Answer: Ok, we now refer as "low flow period" the period between August and October (L267).

*L300: You have to recalculate the percentage value. The lowest value you obtain is 826 mm/yr, which is lower than the value for your reference period and thus not 2% (which corresponds to the value of*

*879mm/yr, which you indicated in the previous version manuscript). Also adapt it in L396 and L425.* **Answer:** That's right. We now state that climate change scenarios suggest relatively small changes in mean annual riparian ET (from -4% to +13%) (L297). Moreover, the discussion reads as follows: "The simulated increase in ET induced by the future lengthening of the vegetative period could be higher than the reduction of ET rates during summer, which ultimately could potentially increase annual riparian water use up to 13%" (L390-391).

*L304: The value for scenario RCP 8.5 percentile 0.25 is higher (8.25%).* **Answer:** Right, we have changed it accordingly (L301).

*L306: It is not possible to see this increase of days with ET > 0 mm/d in Figure 5, since you only show the mean ET over the reference period (which never goes below 0 mm/d!) and not the ET for single years of the reference period. Either remove the cross-reference to Figure 5 or consider including some/all single years of the reference period in the figure.* **Answer:** We have removed the cross-reference.

*L326-L327: This is the only sentence in the manuscript where it is still confusing what you considered as the downstream site (because the 10% riparian zone refer to the local drainage area of 4.42 km$^2$, the model output at the downstream site integrates the flow of all the drainage area). I would suggest to rephrase it.* **Answer:** That's correct. We have clarified that model simulations at the downstream site integrate all processes occurring at whole-catchment scale (L323-324).

*L335: Add: at the downstream site.* **Answer:** Ok (L332, L338).

*L365: I would suggest to remove "saturated".* **Answer:** Ok, removed.

*L373: components instead of component.* **Answer:** Ok (L369).

*L378: Maybe remove "projected for later in this century"* **Answer:** Ok, deleted.

*L390: scenario and year (cf. L306)* **Answer:** Ok, changed (L384).

*L419-L420: You should give the same values here as in the abstract (L25-26); additionally consider to remove "dry".* **Answer:** Ok, we now report the whole range of values (i.e, from 5.5 to 8.4%) (L415-416). Also, we have deleted "dry" from the sentence.